# Acetylation of histone H4 lysine 5 and 12 is required for CENP-A deposition into centromeres

Wei-Hao Shang[1], Tetsuya Hori[1], Frederick G. Westhorpe[2], Kristina M. Godek[3], Atsushi Toyoda[4], Sadahiko Misu[5], Norikazu Monma[5], Kazuho Ikeo[5], Christopher W. Carroll[6], Yasunari Takami[7], Asao Fujiyama[4,8], Hiroshi Kimura[9], Aaron F. Straight[2] & Tatsuo Fukagawa[1]

Centromeres are specified epigenetically through the deposition of the centromere-specific histone H3 variant CENP-A. However, how additional epigenetic features are involved in centromere specification is unknown. Here, we find that histone H4 Lys5 and Lys12 acetylation (H4K5ac and H4K12ac) primarily occur within the pre-nucleosomal CENP-A–H4–HJURP (CENP-A chaperone) complex, before centromere deposition. We show that H4K5ac and H4K12ac are mediated by the RbAp46/48–Hat1 complex and that RbAp48-deficient DT40 cells fail to recruit HJURP to centromeres and do not incorporate new CENP-A at centromeres. However, C-terminally-truncated HJURP, that does not bind CENP-A, does localize to centromeres in RbAp48-deficient cells. Acetylation-dead H4 mutations cause mis-localization of the CENP-A–H4 complex to non-centromeric chromatin. Crucially, CENP-A with acetylation-mimetic H4 was assembled specifically into centromeres even in RbAp48-deficient DT40 cells. We conclude that H4K5ac and H4K12ac, mediated by RbAp46/48, facilitates efficient CENP-A deposition into centromeres.

[1] Graduate School of Frontier Biosciences, Osaka University, Suita, Osaka 565-0871, Japan. [2] Department of Biochemistry, Stanford University Medical School, 259 Campus Drive, Beckman B409, Stanford, California 94305, USA. [3] Department of Biochemistry, Geisel School of Medicine, Dartmouth College, HB7200, Hanover, New Hampshire 03755, USA. [4] Comparative Genomics Laboratory, National Institute of Genetics, Mishima, Shizuoka 411-8540, Japan. [5] DNA Data Analysis Laboratory, National Institute of Genetics, Mishima, Shizuoka 411-8540, Japan. [6] Department of Cell Biology, Yale University School of Medicine, SHM C-230, 333 Cedar St., New Haven, Connecticut 06520, USA. [7] Section of Biochemistry and Molecular Biology, Department of Medical Sciences, University of Miyazaki, 5200, Kihara, Kiyotake, Miyazaki 889-1692, Japan. [8] National Institute of Informatics, Hitotsubashi, Chiyoda-ku, Tokyo 101-8430, Japan. [9] Cell Biology Unit, Institute of Innovative Research, Tokyo Institute of Technology, 4259 Nagatsuta-cho, Midori-ku, Yokohama 226-8501, Japan. Correspondence and requests for materials should be addressed to T.F. (email: tatsuofukagawa@gmail.com).

During faithful chromosome segregation, spindle microtubules attach to kinetochores, which form on the centromere region of each chromosome. Incorrect attachment of microtubules to the kinetochore causes chromosome instability. In many organisms, the centromere region is specified at a single position on each chromosome, the location of which does not depend on the DNA sequence, but is instead epigenetically determined by centromeric chromatin[1]. Nucleosomes containing the histone H3 variant CENP-A are a key epigenetic determinant for centromere specification and maintenance, as they are essential for centromere and kinetochore formation[1–7]. Studies in *Drosophila* and human cells have shown that in addition to CENP-A, nucleosomes within centromeric chromatin have distinct post-translational modification patterns[8–10]. It remains unclear whether additional histone marks help to specify the sites of CENP-A assembly, and whether properties of CENP-A nucleosomes in addition to just the presence of CENP-A participate in centromere specification.

To address this question, we examine centromere-specific histone modifications in this study and find that H4K5ac and H4K12ac are enriched at centromeres. Furthermore, we characterize the functional significance of these modifications to the process of centromere maintenance and conclude that H4K5ac and H4K12ac, mediated by RbAp46/48, are essential for CENP-A deposition through centromere recognition activity of HJURP.

## Results

**Acetylation of histone H4K5 and K12 is enriched at centromeres**. Chicken DT40 cells have at least three non-repetitive centromeres (Chromosome Z, 5 and 27)[11], making it possible to evaluate the coincidence of histone modification-profiles with CENP-A in non-repetitive centromeres (Fig. 1a; Supplementary Fig. 1A,B). Using this strategy, we previously found that H4K20me1 in centromeric chromatin is crucial for kinetochore assembly[12]. In this study, using monoclonal antibodies against various histone H4 modifications (a list in Supplementary Table 1)[13], we used ChIP-seq to identify additional centromere-specific histone H4 modifications. We found that histone H4K5ac and H4K12ac were both enriched at centromere regions in chicken DT40 cells (Fig. 1a–c; Supplementary Fig. 1A,B). Acetylation of histone H4 N-terminal tail lysine residues are predominantly associated with euchromatin, and contribute to chromatin decondensation and transcriptional regulation[14]. We therefore predicted that H4K5ac and H4K12ac must occur at multiple loci beyond centromere regions in the chicken genome. Consistent with this idea, significant accumulation of both H4K5ac and H4K12ac was detected in multiple positions (Fig. 1b,c); if we mapped sequence reads of ChIP samples using H4K5ac and H4K12ac antibodies to the chicken reference genome in 100 kb windows, it was hard to detect clear centromeric peak (Fig. 1b,c, middle panel). However, H4K5ac and H4K12ac ChIP-seq peaks at centromeres were detected after aligning the ChIP-seq profile of H4K5ac, H4K12ac and CENP-A at non-repetitive centromeres in 10 kb windows (Fig. 1b,c, bottom panels). ChIP-seq mapping at a high resolution clearly indicates coincidence of CENP-A with H4K5ac or H4K12ac (Fig. 1a; Supplementary Fig. 1A,B). In contrast, other histone H4 acetylation sites, including H4K8ac, H4K16ac and H4K20ac, were not detected at centromeres even in high resolution (Fig. 1a; Supplementary Fig. 1A,B). Thus, of the acetylation events tested, we conclude that only H4K5ac and H4K12ac are enriched at centromere regions in DT40 cells.

To examine whether H4K5ac and H4K12ac are enriched at human centromeres we performed immunofluorescence analysis using anti-H4K5ac and H4K12ac antibodies in HeLa cells. When we stained HeLa cells expressing CENP-A-green fluorescent protein (GFP) with directly Cy3-labelled H4K5ac or H4K12ac antibodies, signals were observed throughout the entire nucleus (Fig. 1d), supporting our ChIP-seq observations in which these modifications occur at multiple genome regions. However, we could detect enrichment of H4K12ac staining at centromeres, marked with GFP-CENP-A (Fig. 1d, bottom). In addition, we observed co-detection of H4K12ac and endogenous CENP-A in HeLa cells (Supplementary Fig. 1C), but, some H4K12ac signals were weak, which may be due to antibody accessibility (Supplementary Fig. 1C). We did not detect centromere signals for H4K5ac by immunofluorescence. In addition to immunofluorescence analysis, we immunoprecipitated CENP-A from HeLa cells and could detect H4K5ac and H4K12ac by western blot (Fig. 2), suggesting that H4K5ac and H4K12ac are also enriched in human centromeres. In summary, based on ChIP-seq and immunofluorescence analyses we conclude that H4K5ac and H4K12ac are enriched at centromere regions in both chicken and human cells.

**H4K5ac and K12ac occur in the pre-nucleosomal CENP-A–H4**. Newly synthesized H4 is acetylated at K5 and K12 residues in the pre-deposition histone H3 complexes[15]. Therefore, we hypothesized that H4K5 and H4K12 are predominantly acetylated in the pre-nucleosomal CENP-A–H4 complex and these acetylations are reduced upon CENP-A–H4 deposition into centromeric chromatin. To examine these hypotheses, we prepared both the pre-nucleosomal CENP-A–H4 complex and centromeric chromatin containing CENP-A from chicken DT40 cells (Fig. 2a). As only the pre-nucleosomal CENP-A–H4 complex is associated with the chaperone HJURP (refs 16,17), we purified the pre-nucleosomal CENP-A–H4 complex using an anti-FLAG antibody immunoprecipitation of DT40 cells expressing HJURP-FLAG (1st IP). We eluted the FLAG immunoprecipitates and sequentially performed IP with anti-CENP-A antibody (2nd IP) to prepare the pre-nucleosomal CENP-A–H4–HJURP complex. We did not detect any histone H3 in this fraction, confirming this procedure yields purified CENP-A–H4–HJURP complex (Fig. 2b). We also prepared the CENP-A–H4 chromatin fraction by immunoprecipitating with anti-CENP-A antibody from the chromatin pellet after micrococcal nuclease (MNase) digestion. Western blot analysis with antibodies against six different H4 modifications demonstrated that H4K5ac and H4K12ac were the only modifications we could detect in the pre-nucleosomal CENP-A–H4–HJURP complex (Fig. 2b; Supplementary Fig. 1A). We quantitatively compared H4K5ac and H4K12ac levels between the pre-nucleosomal and chromatin CENP-A fractions (Fig. 2c). Although we detected H4K5ac and H4K12ac in both CENP-A fractions, the level of acetylation within chromatin was 30% of that observed in the pre-nucleosomal CENP-A–H4–HJURP complex (Fig. 2c,d). In contrast, H4K20me1 mainly associated with the chromatin CENP-A fraction rather than the pre-nucleosomal CENP-A fraction (Fig. 2c), consistent with our previous results[12].

Consistent with our results in DT40 cells, similar fractionation in HeLa cells (Supplementary Fig. 2B) showed that H4K5ac and H4K12ac predominantly occurred in the pre-nucleosomal CENP-A–H4–HJURP complex (Fig. 2e,f), although a greater proportion of H4K12ac was present in human CENP-A chromatin compared with chicken. This result is consistent with a recent proteomics analysis[18].

In addition, we prepared histones *in vitro* and compared H3–H4 tetramers, CENP-A–H4 tetramers and octameric nucleosomes as substrates for acetylation using histone acetyltransferase 1 (Hat1). *In vitro*, Hat1 preferentially acetylated H3–H4 or CENP-A–H4 tetramers rather than nucleosomes (Supplementary Fig. 2C,D). Taken together, we

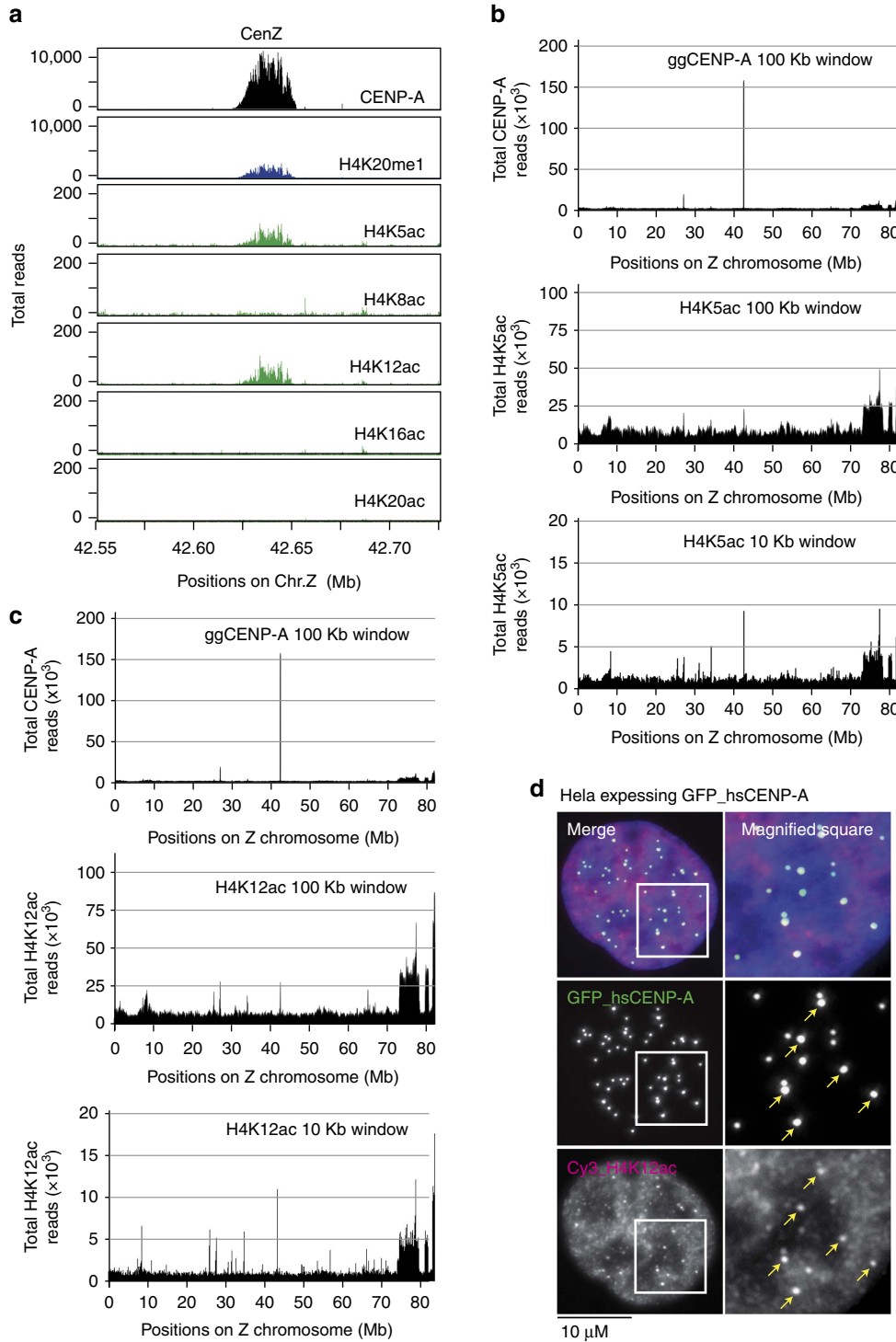

**Figure 1 | H4K5 and K12 acetylation are detected in centromeres.** (**a**) High-resolution profile of ChIP-seq analysis with anti-CENP-A, anti-H4K20me1 or various antibodies against H4 modifications including K5ac, K8ac, K12ac, K16ac and K20ac around centromere region of chromosome Z (42.55–42.725 Mb). (**b**) ChIP-seq analysis with anti-CENP-A or anti-H4K5ac antibodies on chromosome Z in DT40 cells. Sequence reads were mapped for CENP-A at 100 kb window and for H4K5ac at 100 kb and 10 kb windows. At 10 kb windows a peak for H4K5ac at centromere position are clearer. (**c**) ChIP-seq analysis with anti-CENP-A or anti-H4K12ac antibodies on chromosome Z in DT40 cells. Sequence reads were mapped for CENP-A at 100 kb window and for H4K12ac at 100 kb and 10 kb windows. At 10 kb windows a peak for H4K12ac at centromere position are clearer. (**d**) Immunofluorescence analysis with Cy3-labelled anti-H4K12ac antibody (red) in HeLa cells expressing CENP-A-GFP (green). Co-localization of H4K12ac with CENP-A was observed (merge). Typical centromere signals are shown in yellow arrows. Bar, 10 μm.

conclude that H4K5ac and H4K12ac primarily occur in the pre-nucleosomal CENP-A–H4 fraction rather than centromeric chromatin in both chicken and human cells, and that the presence of H4K5ac and H4K12ac in chromatin is likely a consequence of assembly of acetylated pre-nucleosomal CENP-A–H4.

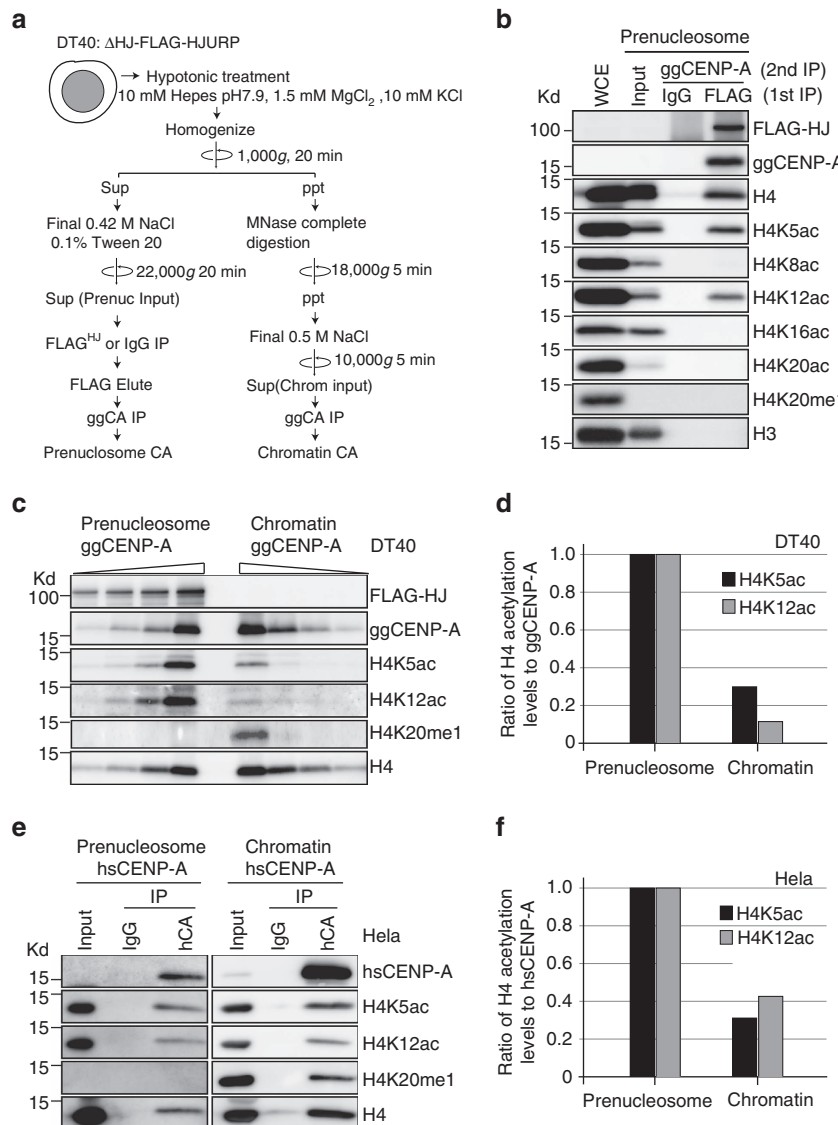

**Figure 2 | H4K5 and K12 acetylation primarily occur in the pre-nucleosomal CENP-A–H4 complex.** (**a**) Experimental strategy for preparation of the pre-nucleosomal CENP-A–H4 complex and CENP-A containing chromatin fractions. To highly purify the pre-nucleosomal CENP-A–H4 complex, HJURP associated fraction was used through IP with anti-FLAG antibody in HJURP-deficient DT40 cells expressing FLAG-HJURP (ΔHJ-FLAG-HJURP). To prepare chromatin fraction nuclear pellet was digested with MNase at low salt condition (90 mM NaCl) and was solubilized in 500 mM NaCl buffer. Then, immunoprecipitation with anti-CENP-A was performed to obtain chromatin CENP-A. (**b**) Western blot analysis on the pre-nucleosomal CENP-A–H4 complex with anti-FLAG, anti-CENP-A, anti-H3 or various antibodies against H4 modifications including K5ac, K8ac, K12ac, K16ac, K20ac and K20me1 in DT40 cells. (**c**) Comparison of levels for H4K5ac, H4K12ac and H4K20me1 in the pre-nucleosomal CENP-A–H4 complex with those in CENP-A containing chromatin fraction. ΔHJ-FLAG-HJURP cells were used for sample preparation. H4 and CENP-A were used for loading control. (**d**) Quantification of levels of H4K5ac and H4K12ac in the pre-nucleosomal CENP-A–H4 complex and CENP-A containing chromatin. Band intensities for H4K5ac and H4K12ac in **c** were normalized to CENP-A levels. (**e**) Comparison of levels for H4K5ac, H4K12ac and H4K20me1 in the pre-nucleosomal CENP-A–H4 complex with those in CENP-A containing chromatin fraction in human HeLa cells. (**f**) Quantification of levels of H4K5ac and H4K12ac in the pre-nucleosomal CENP-A–H4 complex and CENP-A containing chromatin. Band intensities for H4K5ac and H4K12ac in **e** were normalized to CENP-A levels.

**The RbAp46/48–Hat1 complex acetylates H4K5 and K12**. As acetylation of CENP-A–H4 tetramers is mediated by Hat1 *in vitro*, and Hat1 associates with RbAp46/48, the homologue of fission yeast Mis16 (ref. 19), we probed the role of RbAp46/48 in H4 acetylation. We previously created conditional knockout DT40 cell lines for RbAp48 (ref. 20). Chicken possesses RbAp46 and RbAp48, but RbAp46 is not expressed in DT40 cells[20]; therefore, RbAp48-deficient cells express neither RbAp46 nor RbAp48. We also tested H4 acetylation in Mis18α-deficient cells[21]. As both RbAp48 and Mis18α are essential for cell viability, gene expression of each protein is conditionally turned off upon

tetracycline addition. So, we refer to control cells as RbAp48 ON or Mis18α ON cells and knockout cells (after tetracycline addition) as RbAp48 OFF or Mis18α OFF cells. We prepared the pre-nucleosomal CENP-A complex from RbAp48 ON or OFF cells and examined levels of H4K5ac and H4K12ac by western blot. H4K5ac and H4K12ac levels in RbAp48 OFF cells were decreased to <20% of those of RbAp48 ON cells (Fig. 3a,c). In contrast, H4K5ac and H4K12ac were not changed in Mis18α-OFF cells (Fig. 3b,c).

While H4K5ac and H4K12ac were reduced at the CENP-A–H4 complex in RbAp48 OFF cells, we did not detect a change in

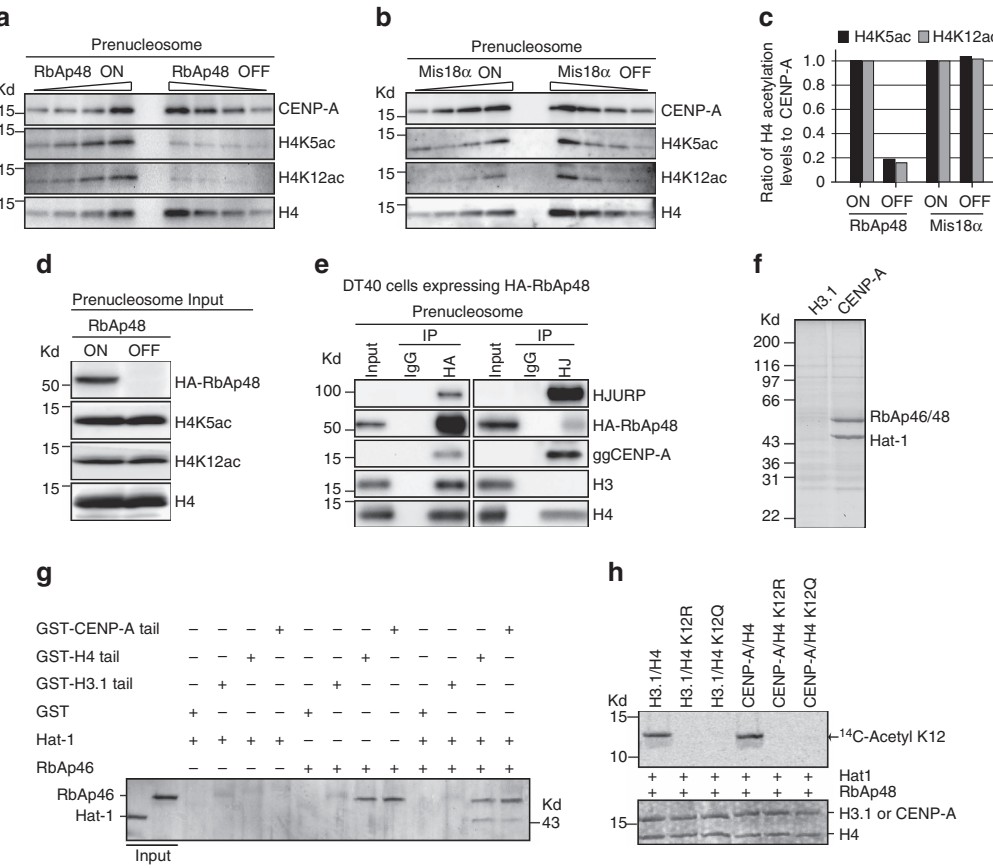

**Figure 3 | H4K5 and K12 acetylation in the pre-nucleosomal CENP-A–H4 complex are mediated by the RbAp48 complex.** (**a**) Comparison of levels for H4K5ac or H4K12ac in the pre-nucleosomal CENP-A–H4 complex in RbAp48 ON cells with those in RbAp48 OFF cells. (**b**) Comparison of levels for H4K5ac or H4K12ac in the pre-nucleosomal CENP-A–H4 complex in Mis18α ON cells with those in Mis18α OFF cells. (**c**) Quantification of levels of H4K5ac or H4K12ac in the pre-nucleosomal CENP-A–H4 complex in RbAp48 ON/OFF or Mis18α ON/OFF cells. Band intensities for H4K5ac and H4K12ac in **a** and **b** were normalized to CENP-A levels. (**d**) Comparison of levels for H4K5ac or H4K12ac in total pre-nucleosomal fraction in RbAp48 ON cells with those in RbAp48 OFF cells. (**e**) Immunoprecipitation with anti-HA or anti-HJURP antibodies in DT40 cells expressing HA-RbAp48, followed by western blot analysis with anti-HJURP, anti-HA, anti-CENP-A, anti-H3 and anti-H4 antibodies. (**f**) Affinity chromatography with histone H3.1 and CENP-A N-termini. *Xenopus* RbAp46/48 and xHat1 bind to the xCENP-A N-terminus. (**g**) xRbAp46 binds directly to the xCENP-A N-terminal tail and xHat1 depends on xRbAp46 for xCENP-A association. Tail fusions to GST and presence of Hat1 or RbAp46 is indicated on the left. Input Hat1 and RbAp46 proteins are indicated. (**h**) The xHat1–xRbAp48 complex acetylates xCENP-A–H4 tetramers on H4K12. Acetylation reactions were performed in the presence of Hat1 and RbAp48. Mutation of H4K12 eliminated detectable [14]C-acetylation of H4 by Hat1. The top panel is an autoradiogram detecting the [14]C-acetylation of histone substrates and the bottom panel is a coomassie stain of the gel.

overall levels of H4K5ac and H4K12ac (Fig. 3d). It is possible that Hat1 is also associated with complexes other than RbAp48, which may be responsible for acetylation away from the CENP-A–H4 pre-nucleosomal complex. Immunoprecipitation experiments confirmed that RbAp48 associates with pre-nucleosomal CENP-A–H4 complex in DT40 cells, suggesting an RbAp48 containing complex acetylates the CENP-A–H4 prenucleosomal complex (Fig. 3e).

RbAp46/48 are known to interact with the amino terminus of histone H4 but not histone H3 (ref. 15). We used purified recombinant amino termini of histone H3.1 and CENP-A to identify proteins in *Xenopus* egg extract that selectively interact with CENP-A but not H3.1. We found that the RbAp46/48–Hat1 complex bound specifically to the CENP-A N-terminus but not to the H3.1 N-terminus (Fig. 3f). We tested whether this interaction was direct by mixing soluble recombinant RbAp46, Hat1 or the RbAp46–Hat1 complex with recombinant glutathione *S*-transferase (GST)-CENP-A or H3.1 N-terminal tails. We found that the CENP-A N-terminus but not the H3 N-terminus precipitated RbAp46 and Hat1, and that this interaction required

RbAp46 because no Hat1 was recovered in the absence of RbAp46 (Fig. 3g). In similar experiments, we find that RbAp48 also binds to the CENP-A N-terminus (Supplementary Fig. 2E). We tested whether RbAp48–Hat1 acetylates CENP-A, H4 or both and found that the acetylation reaction was specific to histone H4 but not CENP-A (Fig. 3h; Supplementary Fig. 2F). By mutating H4K12 to arginine or glutamine we found that lysine 12 is the predominant site of modification *in vitro* (Fig. 3h; Supplementary Fig. 2F). This result is consistent with the preference for K12 by the yeast, human and *Xenopus* Hat1 enzyme[22–27]. We were able to detect a low level of H4K5 acetylation upon long exposure of our acetyltransferase assays (Supplementary Fig. 2F); thus both H4K5 and K12 are acetylated by Hat1 *in vitro* with a strong preference for H4K12. Recently, Ohzeki *et al.*[28] found that Kat7 acetylates centromeric chromatin to facilitate CENP-A assembly. We prepared Kat7-deficient cells, but deleting Kat7 did not cause reduction of H4K5ac and H4K12ac at centromeres (Supplementary Fig. 2G). Taken together, we conclude that H4 acetylation of the pre-nucleosomal CENP-A complex is mediated by the RbAp46/48–Hat1 complex.

**CENP-A deposition is compromised in RbAp48-deficient cells.** We examined CENP-A levels at centromeres in RbAp48 OFF DT40 cells by immunofluorescence with anti-CENP-A antibody. Consistent with previous RNAi-based data in human cells[19], CENP-A levels at the centromeres of RbAp48 OFF cells were reduced to ~40% of control cells (Fig. 4a,b). In parallel with this observation, total CENP-A level was reduced (Supplementary Fig. 3A), which is consistent with previous observations[17,19]. Despite this, levels of the centromere proteins CENP-C and CENP-T were not changed (Fig. 4b; Supplementary Fig. 3B,C). Although CENP-C or CENP-T require CENP-A for their localization to centromeres[1,29] a 40% reduction of CENP-A may not be sufficient for a reduction of dependent centromere proteins, which is consistent with data in human cells treated with CENP-A RNAi (ref. 30) or in which the CENP-A gene has been deleted[31].

To examine why CENP-A levels were reduced in RbAp48 OFF DT40 cells, we tested whether newly synthesized CENP-A was deposited into centromeres using the SNAP-tag assay[32]. As observed in human cells, newly synthesized CENP-A was deposited into G1 centromeres in RbAp48 ON DT40 cells (Fig. 4c). By contrast, incorporation efficiency of newly synthesized CENP-A was dramatically reduced in RbAp48 OFF cells (Fig. 4c–e; Supplementary Fig. 3D). We performed this assay during time period from 44 to 48 h after tetracycline addition to RbAp48 conditional knockout cells. S-phase defect occurs in RbAp48-deficient cells[20], however, as cells are still growing and significant numbers of G1 cells are detected, we did the assay in the time period. Although we prefer to explain that RbAp48 deletion directly causes a defect in CENP-A deposition during G1, we cannot completely rule out the possibility that S-phase defect indirectly causes decrease of CENP-A deposition in G1 cells. We also observed an increase of mis-localized CENP-A in non-centromere regions in RbAp48 OFF cells (Supplementary Fig. 3E–G). In addition, we confirmed the increase of non-centromeric CENP-A in RbAp48-deficient cells using ChIP-seq analysis with anti-CENP-A antibody (Supplementary Fig. 3H). These results suggest that RbAp48 mediates specific incorporation of the CENP-A–H4 complex into centromeres.

While RbAp48 associates with CENP-A and Hat1, it also forms other complexes, including with H3–H4, and functions in several chromatin remodelling complexes[15]. Therefore, it is possible that reduction of CENP-A deposition in RbAp48-deficient cells may be from indirect effects. To address this issue we identified a mutation in RbAp48 that causes a specific reduction of CENP-A binding rather than a full knockout of RbAp48. As a mutation site of fission yeast Mis16 is conserved[19,33] (Supplementary Fig. 4A), we introduced a mutation at the same residue (Y32H) of chicken RbAp48 and replaced wild-type protein with the mutant RbAp48(Y32H) in DT40 cells. We found that the mutant RbAp48 reduced binding to CENP-A (Fig. 4f). Importantly, H4K5ac and H4K12ac were reduced by half in RbAp48(Y32H) expressing mutant cells (Fig. 4g). Consistent with these results, centromere localized CENP-A was reduced in these cells (58% compared with that in control cells) (Supplementary Fig. 4B–E). Finally, we examined the assembly of newly synthesized CENP-A into centromeres based on the SNAP-CENP-A assay in RbAp48(Y32H) expressing mutant cells. Consistent with the effect in the RbAp48-deficient cells, in RbAp48(Y32H) expressing cells new CENP-A deposition was reduced to 63% of control cells (Supplementary Fig. 4F). These data suggest that RbAp48 directly associates with CENP-A and this association contributes to acetylation of the CENP-A–H4 complex and CENP-A deposition into centromeres.

**HJURP localization is impaired in RbAp48-deficient cells.** To address why newly synthesized CENP-A was not incorporated into centromeres in RbAp48 mutant cells, we examined the localization of components that control the assembly of CENP-A, including the Mis18 complex proteins and HJURP. We found that HJURP, but not Mis18 complex proteins, mis-localized in RbAp48 OFF cells (Fig. 5a; Supplementary Fig. 5A–D). Approximately 80% of RbAp48 OFF cells in G1 (when HJURP normally localizes to centromeres[16,17]) showed defective HJURP centromere localization (Fig. 5b,c; Supplementary Fig. 5E). These data show that RbAp48 is required for HJURP recruitment to centromeres. We confirm that the total level of HJURP protein is unchanged in RbAp48 OFF cells (Supplementary Fig. 5F). We also found impaired HJURP recruitment to centromeres in cells expressing the RbAp48 (Y32H) mutant (Supplementary Fig. 4G).

The CENP-A binding domain and centromere-targeting domain of HJURP are distinct[21,34]. For chicken HJURP, the central region (255–571 aa) is essential and sufficient for centromere localization of HJURP, while the N-terminal region (1–254 aa) is critical for CENP-A binding (Fig. 5d). We observed that CENP-A binds to HJURP even in RbAp48 OFF cells (Supplementary Fig. 5G). However, H4 acetylation at K5 and K12 does not properly occur at the CENP-A–H4–HJURP complex without RbAp48 (Fig. 3a). These data suggest that non-acetylated CENP-A–H4 bound to HJURP's N-terminus (1–254 aa) may interfere with centromere recognition via the HJURP middle region (255–571 aa). Alternatively, RbAp48 may be directly involved in recruiting HJURP to centromeres. To examine these possibilities, we introduced an N-terminal truncated mutant for HJURP (HJΔCA) into RbAp48 OFF cells. HJΔCA cannot bind a H4–CENP-A dimer. Strikingly, in contrast to full-length HJURP, HJΔCA localization to centromeres was not lost after deletion of RbAp48 (Fig. 5e). Quantification of HJΔCA signal indicated no change in centromere localization in G1 cells (Fig. 5f). Taken together, these data suggest that the HJURP bound to non-acetylated H4–CENP-A complex does not target to centromeres and that RbAp48 facilitates pre-nucleosomal H4–CENP-A–HJURP targeting to centromeres by acetylating H4K5 and H4K12.

**Non-acetylated H4s do not properly localize to centromeres.** To examine these possibilities, we directly tested the significance of H4 acetylation for CENP-A–H4 deposition by preparing H4K5 and K12 mutants, substituted with either alanine, which lacks a charged group on the amino acid sidechain, or arginine, which maintains the charge of lysine but cannot be acetylated (H4_A5A12 or H4_R5R12 mutants). We conditionally expressed Myc-tagged H4 mutants in DT40 cells expressing SNAP-CENP-A. In this system the expression level of mutant H4 is similar to endogenous H4 (Supplementary Fig. 6A). In cells expressing wild-type Myc-H4 (H4_K5K12) or H4_A5A12, SNAP-CENP-A localized exclusively to centromeres. In contrast, CENP-A mis-incorporation into non-centromere region was frequently observed in cells expressing H4_R5R12 (Fig. 6a,b; Supplementary Fig. 6B), also confirmed by ChIP-seq analysis with anti-CENP-A antibody (Fig. 6c). CENP-A incorporation into non-centromere regions was 25.3% increased in cells expressing H4_R5R12 mutant compared with control cells (Fig. 6c). Consistent with the immunofluorescence microscopy data, CENP-A incorporation into non-centromere regions was not increased in cells expressing H4_A5A12 mutant and wild-type H4_K5K12 (Fig. 6c). Interestingly, a double dose of wild-type H4_K5K12 further reduced non-centromeric CENP-A, suggesting that H4K5ac and H4K12ac might be positively involved in removal of non-centromeric CENP-A, as observed in Drosophila cells[35].

Taken together, we conclude that when H4_R5R12 mutant was expressed, CENP-A mis-incorporation into non-centromere region is increased, a similar phenotype to that observed in

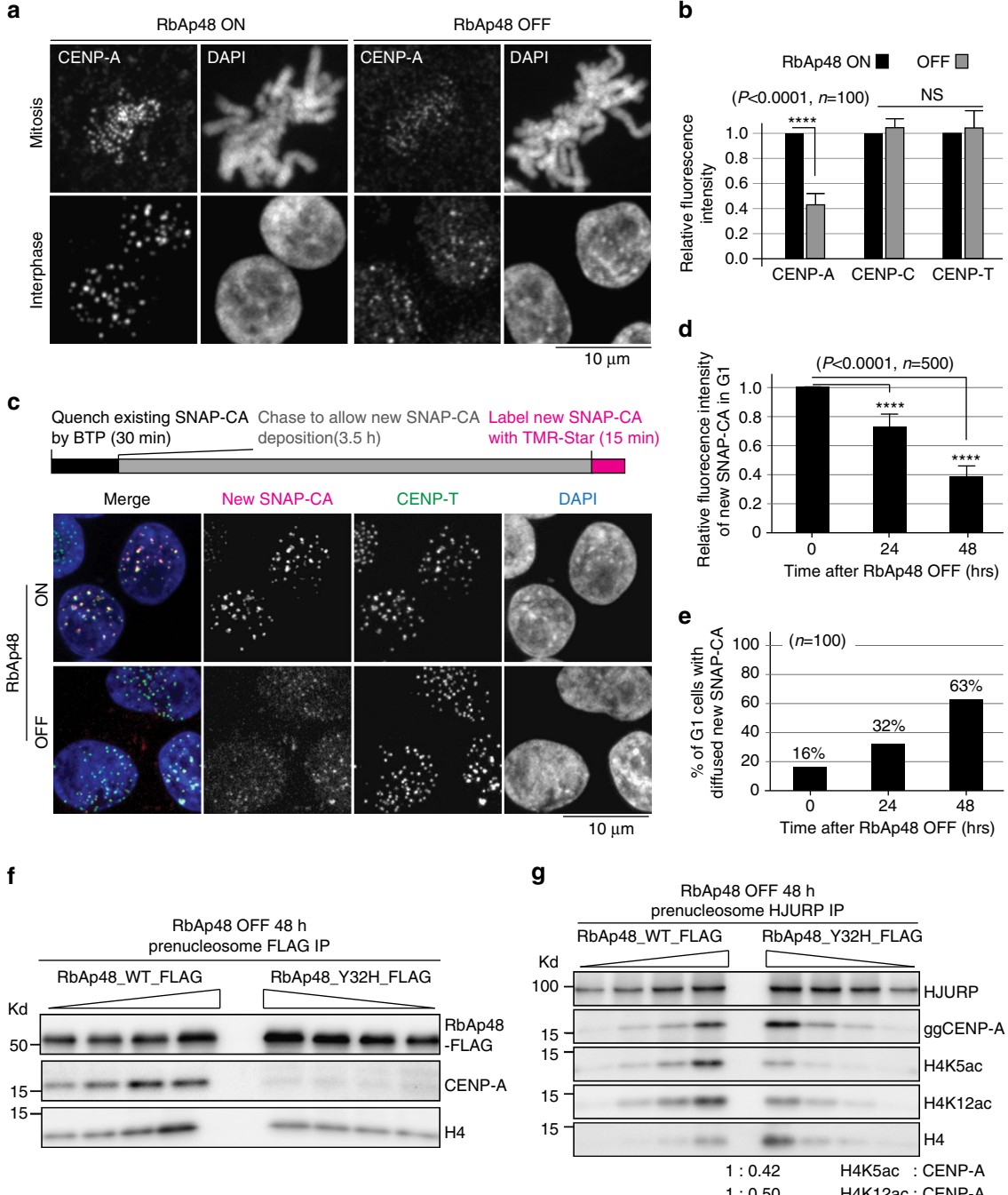

**Figure 4 | CENP-A is not specifically deposited into centromeres in RbAp48-deficient cells.** (**a**) Immunofluorescence with anti-CENP-A antibody in RbAp48 ON and OFF cells. Bar, 10 μm. (**b**) Quantification of levels of CENP-A, CENP-C and CENP-T at kinetochores in RbAp48 ON and OFF cells based on Immunofluorescence analysis. Error bars represent s.d.. Asterisk indicates statistically significance ($P<0.0001$) by Student's $t$-test. ($N=100$). (**c**) Top: outline of quench-chase-pulse experiment in RbAp48 ON or OFF cells stably expressing SNAP-CENP-A. Bottom: representative images of G1 cells in which newly synthesized CENP-A was labelled with TMR-Star in RbAp48 ON or OFF cells. CENP-T was used as a centromere marker. Newly synthesized CENP-A were not detected in RbAp48 OFF cells. Bar, 10 μm. (**d**) Quantification of intensities by TMR-Star at indicated time points after tetracycline addition to RbAp48 conditional knockout cells. Five hundred centromeres in 100 different cells were quantified for each measurement. Error bars represent s.d. Asterisk indicates statistically significance ($P<0.0001$) by Student's $t$-test. ($N=500$). (**e**) Percentages of cells with diffused signals for newly synthesized SNAP-CENP-A are shown at indicated time points after tetracycline addition to RbAp48 conditional knockout cells. Definition of diffusion is in Supplementary Fig. 3D. (**f**) Western blot analysis with anti-CENP-A antibody in immunoprecipitates with anti-FLAG antibody in RbAp48 OFF cells expressing FLAG fused wild-type RbAp48 or Y32H mutant RbAp48. (**g**) Comparison of levels for H4K5ac or H4K12ac in RbAp48 OFF cells expressing FLAG fused wild-type RbAp48 with those in or Y32H mutant RbAp48.

RbAp48 mutant cells. These data suggest that a failure to neutralize lysine charges in the H4 N-terminal tail via acetylation causes mis-incorporation of the CENP-A tetramer.

**Acetylation-mimetic H4 is incorporated into centromeres.** The arginine mutant of H4 (H4_R5R12), which maintains the charge of lysine but cannot be acetylated, causes CENP-A

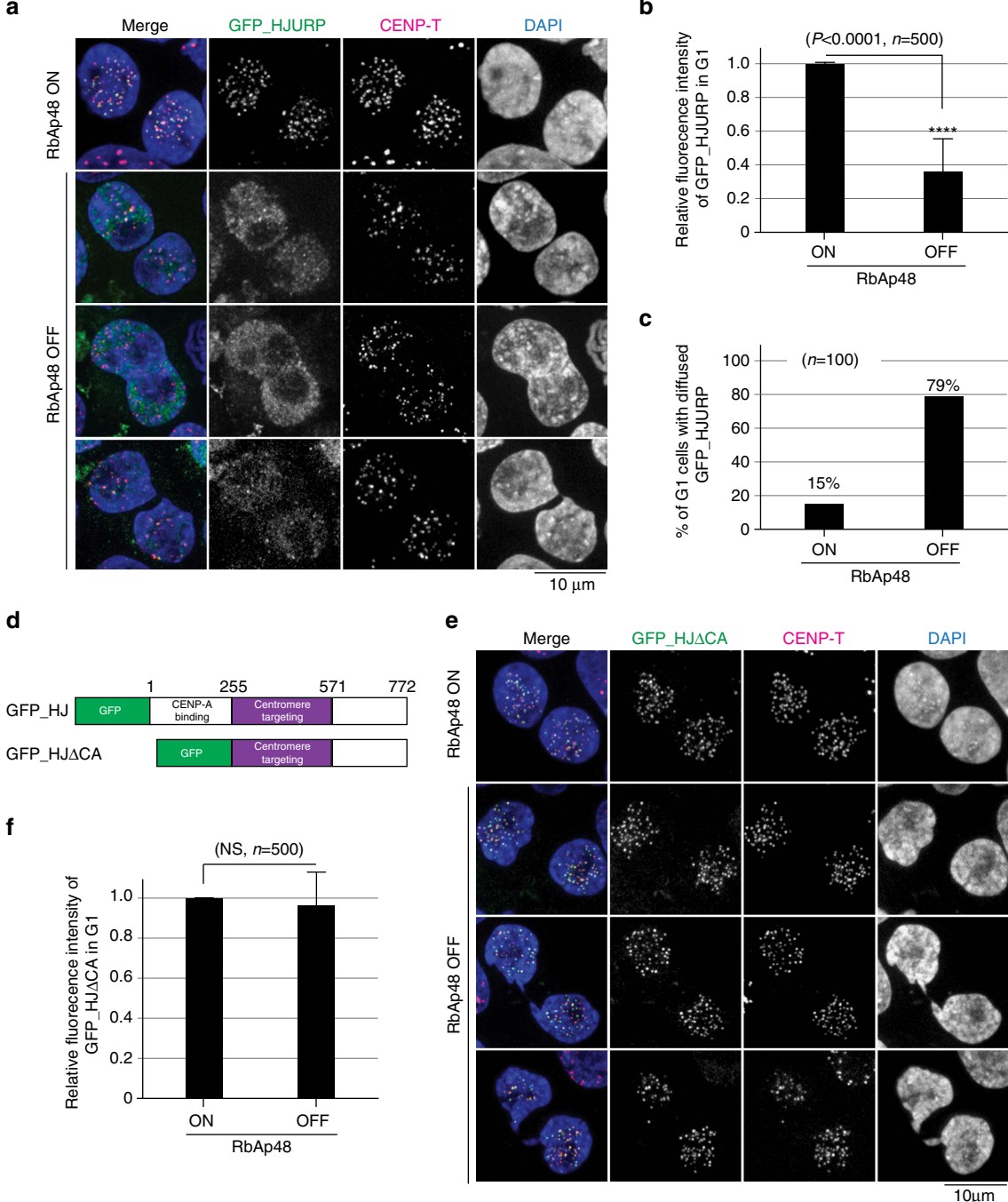

**Figure 5 | HJURP do not properly recognize centromeres in RbAp48-deficient cells.** (**a**) Localization of HJURP in RbAp48 ON or OFF cells stably expressing GFP-HJURP. CENP-T was used as a centromere marker. Bar, 10 μm. (**b**) Quantification of GFP-HJURP intensities at centromeres (shown in **a**) in RbAp48 ON or OFF G1 cells. Five hundred centromeres in 100 different G1 cells were quantified for each measurement. G1 cells were judged by cell size and daughter cell-like morphology. Error bars represent s.d. Asterisk indicates statistically significance ($P < 0.0001$) by Student's $t$-test. ($N = 500$). (**c**) Percentages of cells that displayed diffused GFP-HJURP in RbAp48 ON or OFF G1 cells expressing GFP-HJURP. Definition of diffused GFP-HJURP is in Supplementary Fig. 5E. ($N = 100$). (**d**) Diagram of chicken HJURP protein. The N-terminal region (1–254 aa) is responsible for CENP-A binding. The middle region (255–571 aa) is essential for its centromere localization. We prepared GFP fused full-length HJURP (GFP_HJ) and HJURP lacking CENP-A binding region (GFP_HJΔCA). (**e**) Localization of N-terminal truncated HJURP (GFP_HJΔCA) in RbAp48 ON or OFF cells stably expressing GFP_HJΔCA. CENP-T was used as a centromere marker. Bar, 10 μm. (**f**) Quantification of GFP_HJΔCA intensities at centromeres in RbAp48 ON or OFF cells. Five hundred centromeres in 100 different cells were quantified for each measurement. Error bars represent s.d. of centromere intensity. Relative intensities are shown.

mis-incorporation into non-centromere regions. Considering this result, we hypothesized that positive charge for lysine residues (K5 and K12) in the H4 tail are neutralized by RbAp48–Hat1 complex mediated acetylation, and this process is essential for the specific centromere incorporation of the H4–CENP-A complex.

To test this hypothesis, we stably expressed wild-type H4 (H4_K5K12) or a glutamine mutant of H4 (H4_Q5Q12), which mimics acetylation, in RbAp48 OFF cells and examined CENP-A localization. Expression of H4 mutants does not interfere with tetracycline-mediated repression of RbAp48 (Supplementary

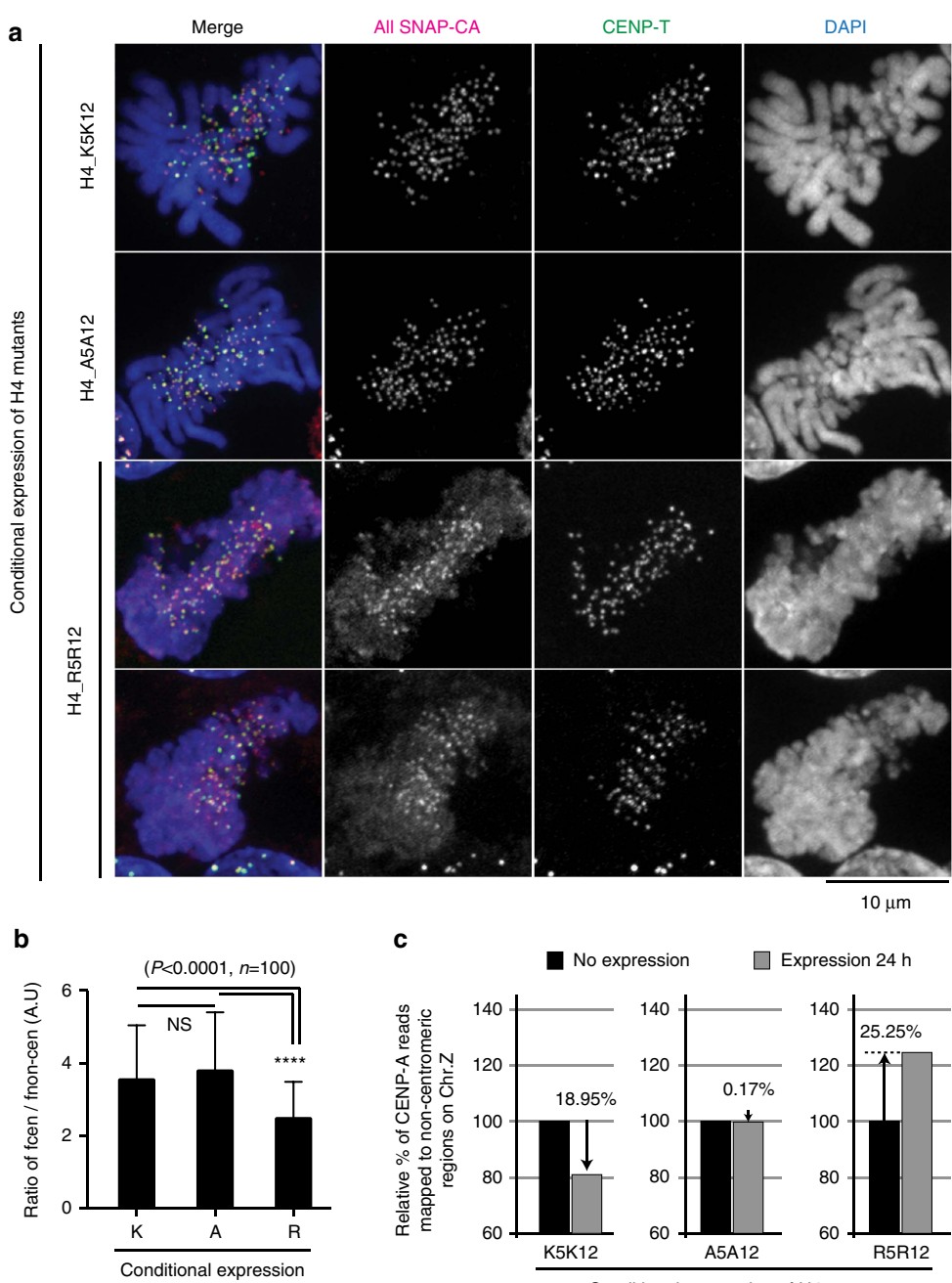

**Figure 6 | Arginine mutation at H4K5 and K12 causes mis-incorporation of the CENP-A–H4 complex into non-centromere regions.** (a) Representative images of SNAP-CENP-A labelled with TMR-Star in DT40 cells conditionally expressing H4_K5K12, H4_A5A12, and H4_R5R12. CENP-T was used as a centromere marker. CENP-A mis-incorporation was especially observed in cells expressing H4_R5R12. Bar, 10 μm. (b) Fluorescent intensity ratio of centromere signals for SNAP-CENP-A to non-centromere signals. When arginine mutant of H4 (H4_R5R12) was expressed, this ratio was lower than in cells expressing H4_K5K12 or H4_A5A12. Error bars represent s.d. Asterisk indicates statistically significance ($P < 0.0001$) by Student's $t$-test. ($N = 100$). (c) Increase of sequence reads in non-centromere region in cells expressing H4_R5R12 based on ChIP-seq analysis with anti-CENP-A antibody.

Fig. 6C). In RbAp48 OFF cells expressing wild-type H4, we observed an increase of CENP-A mis-incorporation into non-centromere regions (Fig. 7a,b), consistent with our analysis of RbAp48-deficient cells (Supplementary Fig. 3E–H). In RbAp48 OFF cells expressing acetylation-mimetic H4_Q5Q12, the ratio of centromeric CENP-A to non-centromeric CENP-A was increased (Fig. 7a,b). In other words, expression of acetylation-mimetic H4 rescues CENP-A mis-incorporation, bypassing the function of RbAp48 in CENP-A centromere assembly.

Finally, we compared the assembly of H4 acetylation mutants into chromatin in *Xenopus* egg extracts containing *in vitro*

reconstituted CENP-A nucleosome arrays as a synthetic substrate (Fig. 7c)[4,36]. While wildtype (H4_K5K12) and acetylation-mimetic H4 (H4_Q5Q12) levels increased after addition of xCENP-A, indicating chromatin incorporation, recruitment of non-acetylatable H4 (H4_R5R12) did not change after xCENP-A addition (Fig. 7d,e). This suggests that unacetylated H4 does not assemble with CENP-A into centromeric chromatin, consistent with DT40 cells expressing H4_R5R12 (Fig. 6a).

Taken together, we conclude that acetylation of H4K5 and H4K12, mediated by the RbAp46/48–Hat1 complex, is essential for CENP-A deposition into centromeres.

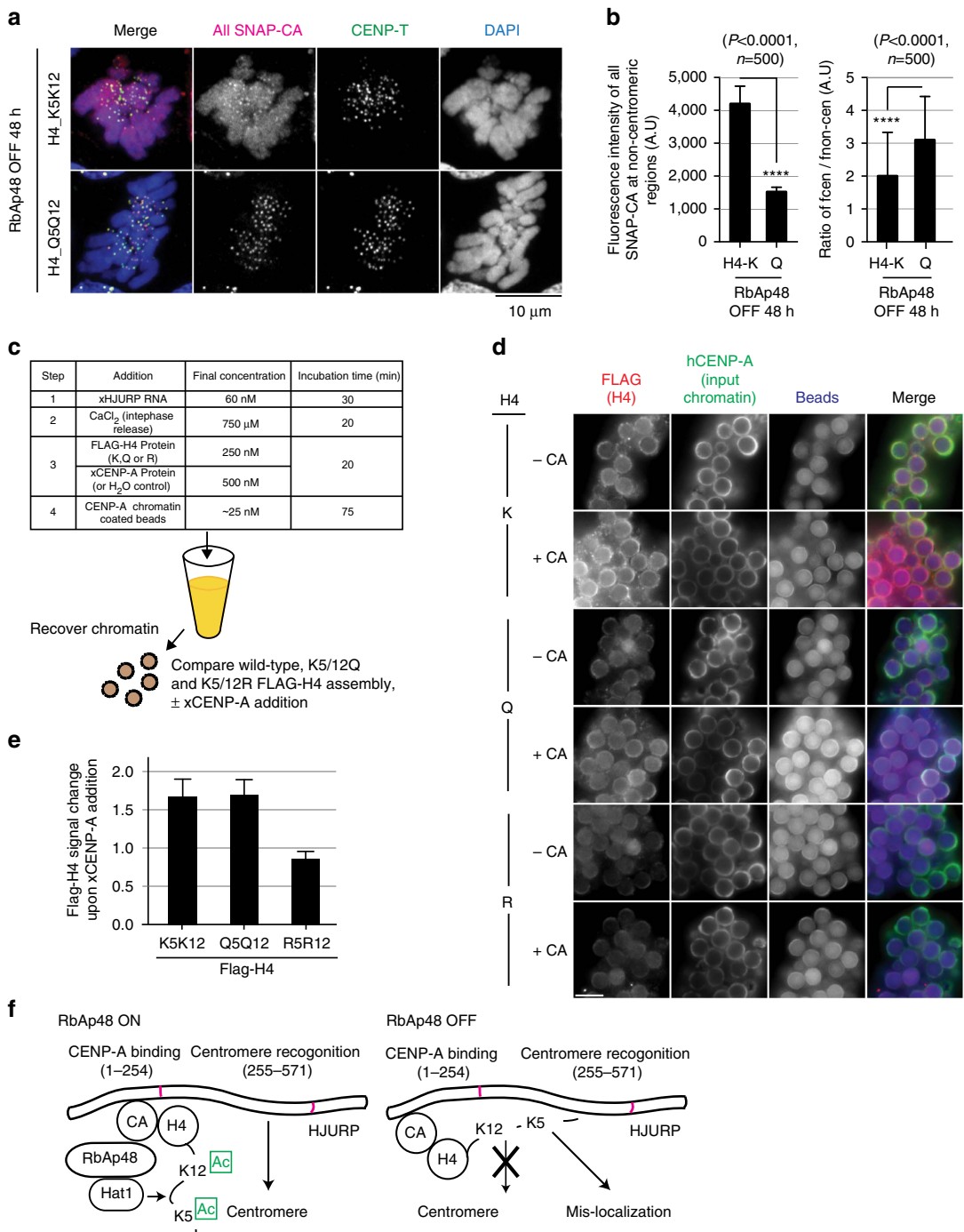

**Figure 7 | Acetylation-mimetic H4Q5Q12 is incorporated into centromeres even in RbAp48-deficient cells. (a)** Representative images of SNAP-CENP-A labelled with TMR-Star in RbAp48 OFF cells expressing H4_K5K12 or H4_Q5Q12. CENP-T was used as a centromere marker. While CENP-A mis-incorporation was observed in RbAp48 OFF cells expressing H4_ K5K12 (top), the mis-incorporation was not observed in RbAp48 OFF cells expressing H4_Q5Q12. Bar, 10 μm. **(b)** Fluorescent intensity of non-centromere signals for SNAP-CENP-A (left) and ratio of centromeric CENP-A to non-centromeric CENP-A (right), when acetyl mimetic H4_Q5Q12 was expressed. Error bars represent s.d. Asterisk indicates statistically significance ($P < 0.0001$) by Student's $t$-test. ($N = 500$). **(c)** Schematic of experimental design to test xCENP-A assembly in the presence of FLAG-tagged histone H4 mutants in *Xenopus* egg extract. Proteins, chromatin beads, RNA or calcium were added to frog egg extract at the concentrations and times indicated in the table. CENP-A chromatin beads are prepared with human CENP-A nucleosomes. After xCENP-A assembly, the chromatin was recovered and assayed for histone assembly using FLAG intensities. **(d)** Fluorescence images of chromatin beads after recovery and immunostaining for human CENP-A to label the input chromatin and FLAG-H4 to label newly incorporated histone H4. **(e)** Levels of histone H4 assembled into chromatin in the presence of CENP-A. **(f)** A proposed role for acetylation of H4 tail mediated by the RbAp46/48-Hat1 complex in the process of CENP-A deposition. The CENP-A–H4 complex is acetylated by the RbAp46/48-Hat1 complex before centromere deposition. The CENP-A–H4 binds the N-terminus of HJURP and middle region of HJURP recognize centromeres for centromere deposition. H4 tail acetylations facilitate this process. However, if the acetylation does not occur properly, non-acetylated tail interferes the centromere recognition for HJURP. So that, CENP-A causes mis-localization into non-centromere regions.

## Discussion

In this study, we conclude that acetylation of H4K5 and H4K12, mediated by the RbAp46/48–Hat1 complex, is essential for CENP-A deposition into centromeres. We propose that positive charge of the lysine residues within the non-acetylated H4 tail interferes with the Mis18 complex:HJURP interaction, possibly through direct interaction of H4 tail to the middle region of HJURP (Fig. 7f). The 255–571 aa region of chicken HJURP is essential and sufficient for centromere localization[21]. Negatively charged aspartate and glutamate residues are enriched in this region (~10%), and it is possible that positive charge of the lysine residues within the non-acetylated H4 tail may directly bind HJURP to interfere with its centromere recognition. The HJURP 255–571 aa region is also serine and threonine rich (more than 20%). A recent report suggested that human HJURP is phosphorylated to facilitate centromere localization[37]. Although the phosphorylation sites in human HJURP are not conserved in chicken, it is possible that chicken HJURP is phosphorylated in the serine and threonine rich region. Thus, one interesting possibility is that the positively charged H4 tail must be neutralized by acetylation. It will be important to clarify the molecular mechanisms through which the acetylated HJURP–H4–CENP-A complex efficiently recognizes centromeres. H4K5ac and H4K12ac stimulate nuclear import through increasing the affinity with Importin 4 (ref. 38). Although it remains unclear how these acetylations contribute to enhancement of affinity with Importin 4, this observation may be related to the efficient incorporation of CENP-A with H4K5ac and H4K12ac into centromeres. Of course, it is possible that additional factors may be involved in the Mis18 complex:HJURP interaction and H4 acetylation may facilitate this process.

Mis16 protein was identified through analysis of a fission yeast Mis16 mutant in which CENP-A incorporation was highly impaired[19]. Similarly, RNAi knockdown for the human Mis16 homologue, RbAp46/48, caused defects for CENP-A incorporation in human cells[17,19]. Biochemical analysis of CENP-A in Drosophila cells identified an interaction of RbAp48 with CENP-A, and depletion of Hat1 from Drosophila cells inhibited CENP-A assembly[39,40]. Caenorhabditis elegans RbAp46/48 homologue LIN-53 also associates with CENP-A and is essential for CENP-A localization, although centromeric function of LIN-53 seems independent histone acetylation[41]. Therefore, RbAp46/48 (Mis16) involvement in CENP-A deposition is evolutionarily conserved. A fission yeast Mis18 mutant displays similar defects in CENP-A deposition[19]. However, it was unclear how the Mis18 complex and Mis16 are functionally related especially in vertebrates. While several studies suggested that Mis16 might form a complex with the Mis18 complex in fission yeast[42,43], RbAp46/48 does not appear to interact with Mis18 in vertebrates[16,17,44]. One current model suggests that the Mis18 complex licences chromatin for the CENP-A deposition as a priming factor and the HJURP–CENP-A–H4 complex recognizes the Mis18 complex[45]. However, it was unclear what role the RbAp46/48 played in the process of CENP-A deposition. We have shown a specific function for RbAp46/48 dependent acetylation of H4K5 and H4K12 that governs HJURP targeting to centromeres for CENP-A assembly (Fig. 7f). Going forward an important question is to understand whether H4 acetylation regulates recognition of the Mis18 complex by the HJURP–CENP-A–H4 complex to control the epigenetic regulation of centromere specification.

## Methods

**Cell culture.** DT40 and HeLa cells were cultured in Dulbecco's modified medium (DMEM) supplemented with 10% fetal calf serum, 1% chicken serum (for DT40), penicillin and streptomycin at 38.5 °C (for DT40) and 37 °C (for HeLa). Plasmid constructs were transfected with a Gene Pulser II electroporator (Bio-Rad) into DT40 cells at 550 V and 25 μF, followed by incubation for 5 min on ice[46]. RbAp48-deficient

cell lines were created in the Takami group[20]. HJURP- and Mis18α-deficient cell lines were created by our group[21]. HeLa cell line was obtained from ATCC.

**ChIP-seq analysis.** For chromatin immunoprecipitation to examine histone modifications we used our collection of antibodies for histone modifications[13,47]. A list of antibodies is shown in Supplementary Table 1. After chromatin was isolated from $1.5 \times 10^9$ cells, chromatin sample was digested with the MNase (Takara), then was extracted with the buffer containing 0.5 M NaCl, and the extract was incubated for 2 h at 4 °C with the Sepharose-Protein G beads, which were pre-incubated with various histone antibodies. Beads were washed and the bound DNA was purified and analysed on HiSeq 2500 DNA sequencer (Illumina). ChIP-seq libraries were constructed according to the Illumina TruSeq DNA LT Sample Prep kit protocols. Approximately 50 ng of purified DNA were end-repaired, followed by the addition of a single adenosine nucleotide at 3′ and ligation to the universal library adaptors. DNA was amplified by eight PCR cycles, and the DNA libraries were prepared. ChIP DNA libraries were sequenced using Illumina HiSeq 2500, in up to $2 \times 151$ cycles[11,48]. Sequenced data were mapped into a Chicken Genome database (NCBI, Build 3.1) with a Burrows–Wheeler Aligner version 0.6.1 programme[49].

**Immunoprecipitation in the pre-nucleosomal CENP-A–H4.** Chromatin-free extracts were prepared from HJURP-deficient DT40 cells ($1 \times 10^9$) expressing FLAG-HJURP. Cells were homogenized with a loose-fit pestle in hypotonic buffer (10 mM HEPES, pH 7.9, 1.5 mM $MgCl_2$, 10 mM KCl, 0.5 mM dithiothreitol (DTT), protease inhibitors). Homogenized extracts were centrifuged at 1,000g for 20 min. The supernatant was solubilized by addition of one tenth volume of Buffer B (120 mM HEPES, pH 7.9, 4.5 M NaCl, 22 mM EDTA, 1.1% Tween20) and centrifuged at 22,000g for 20 min. A total of 20 μg of mouse-IgG or anti-FLAG M2 antibodies at 1:1,000 dilution (F1804, Sigma) conjugated with Dynabeads-M280 (Invitrogen) were added to the supernatant and were incubated for 2 h with rotation at 4 °C. After washing beads at three times in Buffer C (20 mM HEPES, pH 7.9, 0.42 M NaCl, 1.5 mM $MgCl_2$, 0.5 mM EDTA, 0.1% Tween20, 0.5 mM DTT, protease inhibitors), protein complexes were eluted from the beads in Buffer C containing 75 μg of 3xFLAG peptides (F4799, Sigma) for 2 h at 4 °C and processed for the second immunoprecipitation with Dynabeads-protein G conjugated anti-ggCENP-A antibody (1:1,000).

For the chromatin fraction, nuclear pellets were completely digested with MNase (Takara) of 60 unit per ml in Buffer A (15 mM Tris–HCl, pH 7.4, 15 mM NaCl, 60 mM KCl, 1 mM $CaCl_2$, 0.34 M sucrose, 0.5 mM spermidine, 0.15 mM spermine, 1 mM DTT, protease inhibitors) for 1 h at 37 °C. After centrifugation at 18,000g for 5 min, chromatin pellets were re-suspended in a suspension buffer (20 mM Tris–HCl, pH 8.0, 0.5 M NaCl, 10 mM EDTA, protease inhibitors) and centrifuged again at 10,000g for 5 min. The supernatant was subjected to the immuno-precipitation with Dynabeads-proteinG conjugated anti-ggCENP-A or Dynabeads-M280 conjugated anti-HA antibodies (H3663, Sigma) at 1:1,000 dilution.

**SNAP assay.** DT40 cells stably expressing SNAP-CENP-A were quenched by addition of 2 μM bromothenylpteridine (BTP) (SNAP-Cell Block, New England Biolabs) to growth medium for 30 min for irreversible and non-fluorescent labelling of the existing SNAP-CENP-A pool, followed by a 3.5 h chase for new SNAP-CENP-A deposition. Newly deposited SNAP-CENP-A was pulse-labelled with 3 μM of TMR-Star (SNAP-Cell TMR-Star, New England Biolabs). After 15 min, TMR-star labelled cells were washed three times with medium and incubated for 30 min to allow excess TMR-star substrate to be released from cells. Then, cells were fixed with 4% paraformaldehyde (PFA) and performed immunostaining with Alexa488-conjugated anti-CENP-T (1:1,000).

**Immunofluorescence.** For immunofluorescence analysis for H4K12ac in HeLa, cells expressing CENP-A-GFP were treated with hypotonic buffer (20 mM Tris–HCl pH 7.4, 1.5 mM KCl) at room temperature for 10 min and were cytospun into glass-slides. The samples were fixed in 4% paraformaldehyde for 15 min. Then, the samples were treated with 0.5% Triton X-100 in PBS for 5 min and Cy3-labelled mouse monoclonal antibody against H4K12ac was added at 1:1,000 dilution.

Immunofluorescence images for centromere proteins[6] were collected with a Cool SNAP HQ camera (Roper Scientific Japan) mounted on an Olympus IX71 inverted microscope with a ×100 objective lens together with a filter wheel or a CV1000 spinning disk confocal microscope (Yokogawa) with EM CCD. Data analyses were used with Metamorph software (Molecular Devices). Signal intensities were calculated by a method described by Hoffman et al.[50]

**Histone acetyltransferase assays.** All histone acetyltransferase (Hat) assays were done by adding 0.15 μCi [1–$^{14}$C] acetyl coenzyme-A in 0.01 M sodium acetate pH 6.0 (GE-Healthcare, 59 mCi per mmol) to reactions for a final concentration of 166 μM acetyl coenzyme-A. Hat-1 was added to Hat reactions at a final concentration of 30 nM, and RbAp46 or RbAp48 was added at a final concentration of 130 nM. For the Hat reactions including histone tetramers, histone dimers and mononucleosome substrates, the final concentration of all histone substrates in the reactions was 0.48 μM. In the Hat reactions testing only histone tetramer substrates, the final concentration of tetramers was 2.5 μM.

All Hat reactions were performed in 50 mM Tris–HCl pH 8.0, 50 mM KCl, 5% glycerol, 0.1 mM EDTA, 1 mM DTT, 1 mM phenylmethylsulphonyl fluoride (PMSF), and 10 mM sodium butyrate (Sigma). The reactions were assembled on ice and acetyl coenzyme-A was added last. The reactions were incubated at 30 °C for 30 min and stopped with the addition of 4× sample buffer (10% SDS, 50% glycerol, 0.05% bromophenol blue, 0.2 M Tris–HCl pH 6.8, 0.06 M EDTA, and 20% β-mercaptoethanol). The samples were run on 20% SDS–polyacrylamide gel electrophoresis (PAGE) gels, stained with coomassie blue, and destained. The gels were subsequently dried for 1 h at 80 °C. The acetylated products were visualized by exposure of a phosphorscreen (GE-Healthcare) to the gel and scanning of the phosphorscreen on a Typhoon imager (GE-Healthcare).

**Histone tail binding experiments.** Histone tails covalently coupled to NHS Sepharose were washed in 20 mM K + HEPES pH 7.4, 50 mM KCl, 0.1% Tween-20 and 5 mM DTT binding buffer. Each pull-down reaction used 20 μl of histone tail resin with 0.7 μg Hat1, RbAp46 or RbAp48 mixed in 150 μl binding buffer. Beads were incubated for 2 h at 4 °C and then washed four times with 400 μl of 20 mM K + HEPES pH 7.4, 300 mM KCl, 0.1% Tween-20 and 5 mM DTT. After the last wash, the beads were resuspended in 30 μl 4× sample buffer. The samples were run on a 12.5% SDS–PAGE gel and stained with coomassie blue.

***Xenopus* extract chromatin assembly assay.** *Xenopus* extracts were first supplemented with xHJURP RNA, produced by SP6 Message Machine (Thermo Fisher), before being released into interphase with CaCl₂. Recombinant FLAG-H4 and xl CENP-A protein (where indicated) were added to extracts and incubated at 18 °C for 20 min. CENP-A nucleosomal arrays and chromatin coated beads were prepared were assembled on 186 bp of α-satellite DNA[36,51] and incubated in FLAG-H4–CENP-A containing extract for 75 min. Chromatin was recovered from extract and analysed for FLAG-H4 recruitment by immunofluorescence[36].

**DNA constructs.** We generated GST fusions to *Xenopus laevis* histone H3.1, H4 and CENP-A amino termini (amino acids 1–41, 1–42 and 1–50 respectively) in pGEX-6P-1 (GE-Healthcare). *Xenopus* histone H3.1 was cloned into pET3aTr (ref. 52) vector to generate ASP342, codon optimized histone H4 in the pET3a vector was provided by Geeta Narlikar. For the expression and purification of *X. laevis* H2A–H2B dimers, H2A and H2B were cloned into pET3aTr to yield ASP340 and ASP341, respectively. Histones H2B and H2A were then cloned into the polycistronic vector pST39 (ref. 52) generating ASP347. The cloning of untagged CENP-A–H4 tetramers was done by a standard protocol[51].

**Protein expression and purification.** To generate the *Xenopus* H3.1 and *Xenopus* CENP-A affinity columns, GST fusions to H3.1 and CENP-A N-termini were expressed in *Escherichia coli* BL21[DE3]pLysS (Novagen) by induction with IPTG for 3 h at 37 °C. Cells were lysed in 1× PBS containing 0.5 M NaCl, 0.5% Igepal-CA630, 5 mM β-mercaptoethanol, 1 mM EDTA, 10 μg per ml each leupeptin, pepstatin, and chymostatin (LPC) and 1 mM PMSF. The cell lysate was clarified by centrifugation at 100,000g for 1 h and purified on glutathione agarose (Sigma). Eluates from the glutathione agarose columns were applied to Q-sepharose (GE-Healthcare) in 10 mM Tris–HCl pH 8.0, 5 mM β-mercaptoethanol, 10% glycerol, and eluted with a 0–0.5 M NaCl gradient. Peak fractions were pooled and dialyzed into 10 mM HEPES pH 8.0, 500 mM NaCl and 10% glycerol. Purified GST fusions were then coupled to Affigel-10 (Bio-Rad) at a ratio of 3.5 mg protein per ml of resin to generate affinity columns. For histone tail binding experiments, purified GST fusions were coupled to NHS-Sepharose (GE-Healthcare) at a ratio of 5 mg protein per ml of resin.

Bacmids for Flag Hat1, Flag RbAp46 and Flag RbAp48 were generated according to the Bac-to-Bac expression system (Thermo-Fisher). Flag Hat1, Flag RbAp48 and Flag RbAp46 baculoviruses for SF9 cell infection were generated according to Fitzgerald et al.[53] For protein expression, we infected SF9 cells with Flag Hat1, Flag RbAp48 or Flag RbAp46 virus for ∼60 h, harvested the cells by centrifugation and flash froze the cell pellets. The pellets were lysed by douncing on ice in 20 mM potassium phosphate pH 7.4, 250 mM NaCl, 1 mM PMSF, 0.5 mM DTT, 1 mM benzamidine hydrochloride, 10 μg per ml LPC, 0.1% Igepal and 0.2 mM EDTA and then clarified by centrifugation in a JA-20 rotor (Beckman) at 12,000 r.p.m. for 10 min at 4 °C. Supernatants were bound to Flag affinity resin (Sigma) for 4 h at 4 °C, washed with wash buffer containing 250 mM NaCl, 20 mM potassium phosphate pH 7.4, 0.2 mM EDTA, 0.1% Igepal and 0.5 mM DTT (Buffer A), in Buffer A containing 0.5 M NaCl followed by washing in Buffer A. We eluted the bound proteins with Buffer A containing 0.4 mg per ml Flag peptide (Sigma) pooled the eluted fractions then dialyzed them overnight at 4 °C in 20 mM Tris–HCl pH 8.0, 50 mM NaCl, 5 mM β-mercaptoethanol and 1 mM EDTA (Buffer B). The dialyzed fractions were purified by binding to a 1 ml Mono-Q column (GE-Healthcare), washing with 120 mM NaCl Buffer B and eluting with a linear gradient from 120 to 800 mM NaCl in Buffer B. The peak fractions of Hat1 or RbAp48 were flash frozen and stored at −80 °C. The peak fractions of RbAp46 were further purified by gel filtration over a Superdex 200 column (GE-Healthcare) in 500 mM NaCl Buffer B. The peak RbAp46 fractions were pooled and concentrated by centrifugation using an Amicon 4 ml 10,000 MWCO concentrator (Millipore).

Untagged H3.1–H4 tetramers, FLAG-H4 (including acetylation mutants) and H2A–H2B dimers were purified by a standard method[54] CENP-A–H4 tetramers were also purified[51]. *Xenopus* CENP-A was purified from a constitutive expression vector, pHCE[55], using an N-terminal 6His tag.

**Affinity chromatography.** Mitotic *Xenopus* extracts were prepared by a standard protocol[56]. Egg extracts (6 ml) were diluted eightfold into affinity column buffer (50 mM HEPES pH 7.7, 100 mM KCl, 1 mM EGTA, 10 mM MgCl₂, 1 mM DTT) containing 1 mM PMSF, an ATP regenerating system (7.5 mM creatine phosphate, 1 mM ATP, 0.1 mM EGTA, and 1 mM MgCl₂) and 1 μM Microcystin-LR. Extracts were clarified for 90 min at 200,000g, and passed through a 25 ml column of immobilized GST to deplete GST binding proteins. After application of extract, affinity columns were washed with 10 column volumes of column buffer and eluted with 100 mM increments of increasing KCl (three column volumes per elution) in affinity column buffer. Eluates were precipitated with 10% trichloroacetic acid.

**Western blot images.** Uncropped gel images used in Figs 2b,c, 3a,b,e and 4f,g are displayed in Supplementary Fig. 7.

**Data availability.** ChIP-seq data in this paper were deposited into DDBJ Database with accession number DRA004753. The data that support the findings of this study are available from the corresponding author upon request.

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

## Acknowledgements

The authors are very grateful to R. Fukuoka, S. Teramoto and Y. Fukagawa for technical assistance. We also thank Lars Jansen for providing us a SNAP-CENP-A construct. This work was supported by MEXT KAKENHI Grant Numbers 25221106 and 15H05972 to TF, MEXT KAKENHI Grant Number 16K07449 to WHS and by NIH R01 GM 074728 to AFS.

## Author contributions

W.-.H.S. performed entire DT40 and HeLa experiments and analysed the data. T.H. contribute to identification of histone modifications in centromeres and performed experiments to evaluate significances of these modifications. F.G.W. performed chromatin assembly assay using *Xenopus* egg extracts. K.M.G. performed biochemical experiments to show relationship of H4K5ac and H4K12ac with RbAp46/48–Hat1 complex under supervision of C.W.C. and A.F.S. A.T. and A.F. performed deep sequencing. S.M., N.M. and K.I. Analysed deep sequencing data. Y.T. created an RbAp48-defeicient DT40 cell line. H.K. generated antibodies for various histone modifications. A.F.S. suggested some experimental designs and contributed to preparation of the manuscript. T.F., W.-H.S. and T.H. organized all experimental designs and T.F. wrote the manuscript.

## Additional information

**Competing financial interests**: The authors declare no competing financial interests.

**How to cite this article**: Shang, W.-H. *et al.* Acetylation of histone H4 Lysine 5 and 12 is required for CENP-A deposition into centromeres. *Nat. Commun.* **7,** 13465 doi: 10.1038/ncomms13465 (2016).

**Publisher's note**: 

