## [Peer Review File · Nature Communications]

Reviewers' comments:

Reviewer #1 (Remarks to the Author):

In this study, the authors show the presence of acetylated histone H4 lysine 5 and 12 in centromeric chromatin and they demonstrate that prenucleosomal CENP-A/H4 complexes bear the same modifications in chicken DT40 and HeLa cells. In further experiments it is shown that acetylation of prenucleosomal H4 in CENP-A/H4 complexes is dependent on the RbAp48-Hat1 acetyltransferase complex and that efficient CENP-A/H4 incorporation at the centromere is dependent on RbAp48. Importantly, Shang et al find that acetylation of H4K5 and K12 is involved in the correct targeting of the HJURP-CENP-A/H4 complex to the centromere and that acetylation-mimetic mutations in H4 can bypass the requirement of RbAp48.

The role of RbAp48 in binding prenucleosomal CENP-A/H4 and facilitating its incorporation into centromeric chromatin is well established (e.g. Furuyama et al PNAS 2006, Hayashi et al., Cell 2004)) and it has also been shown that the RbAp48-Hat1 complex binds to CENP-A/H4 and that H4 is acetylated in this complex (Barth et al., 2014; Boltengagen et al., 2015). The significance of H4 acetylation in this context, however, has not been addressed so far, and the authors provide interesting results regarding this question that can contribute to advance understanding of centromere assembly mechanisms. There are some issues, however, that should be addressed to strengthen their conclusions.

1. In Figure 1D the signals for H4K12ac seem to overlap only with the brightest CENP-A spots but not with less intense ones. Do the authors have an explanation for this? A control staining with an antibody against a mark that is not enriched at centromeres (e.g. H4K16ac) would be helpful.

2. To confirm that the observed reduction in CENP-A signal upon RbAp48 depletion is directly connected to failed CENP-A incorporation and not due to an arrest of cells in S- or G2 phase given that RbAp48 contributes to the activity of several different protein complexes, the authors should show cell cycle profiles of these cells.

3. In Figure 4 it is shown that the Y32H mutation of RbAp48 reduces binding to CENP-A. However, the authors do not demonstrate that this mutation does not interfere with RbAp48's binding to other complexes. If such proof cannot be obtained, Fig. 4H, I and J provide no substantial additional information to what has already been shown before and could be removed to Supplementary Data.

4. As pointed out above and acknowledged by the authors, RbAp48 is a very versatile protein participating in many complexes that are also involved in gene regulation. Therefore, the authors should show that different HJURP expression levels are not the reason for the observed differences in HJURP localization observed in Fig. 5. They do show decreased CENP-A levels in whole cell extracts of RbAp48 OFF cells which is explained by decreased incorporation into chromatin. However, also in this case it needs to be made sure that expression is not affected.

5. The effects of H4 K5 and K12 mutation on CENP-A mislocalization are interesting. However, the authors do not discuss the possibility that this effect could be secondary to an overall perturbed chromatin structure. In fact, the ChIP data show a decrease of CENP-A signal outside the centromere upon expression of wild-type H4, which can be acetylated and is present in the

cells at double dose (shown in Figure S6A). Thus, problems with global efficient reassembly of chromatin during S-phase in cell expressing the acetylation-mutant H4 might promote the previously observed propensity of CENP-A incorporation into extracentromeric chromatin (e.g. Moreno-Moreno et al., NAR 2006).

6. Fig. 2B: Is the figure mislabeled (it says IP with anti FLAG) or did the authors not use tandem IPs to generate prenucleosomal CENP-A extracts as stated in 2A? Also, why are there no HJURP and CENP-A signals in WCE and Input in 2B, when they can readily be detected in all other blots? When solubilizing chromatin with MNase digestion, usually the supernatant after digestion is used. Why did the authors use the pellet in this case (Fig. 1A)?

Finally, the way the manuscript is written it tends to oversell the findings sometimes. For example, in lines 87-88 it says that H4K5ac and K12 ac are "strikingly,highly enriched..", when the ChIP traces do not suggest "high" enrichment. Also, acetylation assays using Hat1 are described on page 5. On page 6 in a new paragraph, the question of which enzyme acetylates H4 K5 and K12 is raised again, in which the Mis18 complex is named as a possible candidate, even though there is no reason to suspect that this complex acts as a Hat enzyme. The authors should streamline these passages.

Reviewer #2 (Remarks to the Author):

This manuscript by Shang et al uncovers a role for the RbAp48 histone chaperone in CENP-A assembly. The first indication of such a role dates back to 2004 and its role has been enigmatic since. This paper sheds the first clear mechanistic light on its function at the centromere and as such this is of great interest.

The authors find that chicken DT40 centromeres are enriched for H4K5 and K12 acetylation coinciding with CENP-A. They find H4K12ac to be enriched at the centromere by microscopy. These modifications are enriched predominantly on prenucleosomal CENP-A.

Prenucleosomal CENP-A interacts with RbAp46/48 and HAT-1 as previously observed but not detectably with H3 (in the absence of H4). The HAT-1 interaction with CENP-A/H4 is RbAp46 dependent. Within the prenucleosomal complex H4K5 and 12 are specifically acetylated.

In cells, in the absence of RbAp48, HJURP is lost from the centromere but only if HJURP can interact with CENP-A. HJURP mutants that lack the ability to bind CENP-A localize normally in the absence of RbAp48. This result is somewhat puzzling and suggest that CENP-A/H4 binding can interfere with HJURP targeting to the centromere, unless bound by RbAp48 and or acetylated on H4.

Finally and importantly, Acetyl mimicking mutations on H4 when force expressed in cells can rescue CENP-A targeting to centromeres.

Overall, this paper makes a compelling case on the role of RbAp48 in centromere assembly in DT40 cells. The authors present a comprehensive set of experiments that are consistent with a model where RbAp48 brings HAT-1 to the CENP-A/H4 prenucleosomal complex to acetylate H4 at K5 and k12 that is required for subsequent HJURP mediated deposition of CENP-A/H4 at the centromere. This finding is certainly a valuable contribution to the field and should be published.

However, I disagree that these acetylation events should be labelled as "epigenetic marks" for centromere maintenance. H4K5Ac and H4K25Ac are very general modifications that occur on virtually all H4 and is required for general chromatin assembly. I therefore strongly disagree with calling these modifications "epigenetic marks for centromere maintenance". They are neither marks, nor specific for the centromere. They are simply required, just as e.g. the ribosome is required for centromere maintenance. What the authors have done is to demonstrate their requirement and elucidated the role of RbAp48 in this process. There is no need to invoke terminology that muddles their findings. The title and text should be modified to better match their data.

Other key issues to be addressed before publication:

Figure 1D:

Why is ectopic CENP-A GFP overexpression performed here? It is possible that H4K12Ac enrichment at the centromere is induced by CENP-A overexpression. Why not use antibody staining against endogenous centromere proteins to mark centromere position?

Figure 3G:

Ref 27 showed that specificity between Mis16 (the RbAp homolog in fission yeast) and CENP-A comes from Mis16 binding Scm3, the fission yeast HJURP homolog. In the reactions in figure 3G there is no HJURP. How do the authors explain the specific interaction of RbAp46 to CENP-A and the apparent species difference?

Line 165:

"Another candidate is the Mis18 complex, as tricostatin A, a histone deacetylase inhibitor, suppresses the phenotype of Mis18 α -knockdown cells".

Masumoto's lab has recently identified a relevant HAT (KAT7) that may be responsible. This work should be cited and discussed here.

Figure 3H, By mutating K12, most of HAT1 mediated acetylation is lost, suggesting that K12 is the main target for HAT1. This does not exclude the possibility that K5 is not a main target as well. For instance, if K5 and K12 are interdependent then lack of K5 acetylation may prevent K12 acetylation and visa versa. To clearly identify K12 as the main target mutant K5 should be tested as well in this assay.

Figure 4F

The Y32H mutant of RbAp48 has impaired CENP-A binding but normal HJURP binding. The authors observe that HJURP targeting to centromeres is prevented in RbAp48 mutants presumably due to a lack of acetylation of H4 which they suggest interferes with HJURP targeting to centromeres.

Alternatively, RbAp48 is required for HJURP to bind to centromeres through a direct interaction as has been suggested by the work in fission yeast (Ref27). If so, this would predict that in the RbAp48 mutant at Y32H, HJURP would still target normally to centromeres, as in this case no unacetylated CENP-A/H4 is present in the HJURP/RbAP48 complex. This should be tested in order to distinguish between these possibilities.

Figure 7B. The loss of non-centromeric CENP-A is shown, why not quantify directly the rescue of centromeric CENP-A levels? It is critical to directly demonstrate and quantify to what extent K5 and K12 mutations rescue the CENP-A loading defect. The authors present a rather indirect measure.

Perhaps mutant H4 interferes with tetracycline mediated repression of RbAp48. Although unlikely, this would be a trivial explanation for the rescue. Did the authors check this RbAp48 loss in the OFF condition in the presence of mutant H4?

Response to Reviewers

Shang et al. (Nature Communications manuscript NCOMMS-16-14845-T)

Reviewer #1

In this study, the authors show the presence of acetylated histone H4 lysine 5 and 12 in centromeric chromatin and they demonstrate that prenucleosomal CENP-A/H4 complexes bear the same modifications in chicken DT40 and HeLa cells. In further experiments it is shown that acetylation of prenucleosomal H4 in CENP-A/H4 complexes is dependent on the RbAp48-Hat1 acetyltransferase complex and that efficient CENP-A/H4 incorporation at the centromere is dependent on RbAp48. Importantly, Shang et al find that acetylation of H4K5 and K12 is involved in the correct targeting of the HJURP-CENP-A/H4 complex to the centromere and that acetylation-mimetic mutations in H4 can bypass the requirement of RbAp48.

The role of RbAp48 in binding prenucleosomal CENP-A/H4 and facilitating its incorporation into centromeric chromatin is well established (e.g. Furuyama et al PNAS 2006, Hayashi et al., Cell 2004)) and it has also been shown that the RbAp48-Hat1 complex binds to CENP-A/H4 and that H4 is acetylated in this complex (Barth et al., 2014; Boltengagen et al., 2015). The significance of H4 acetylation in this context, however, has not been addressed so far, and the authors provide interesting results regarding this question that can contribute to advance understanding of centromere assembly mechanisms. There are some issues, however, that should be addressed to strengthen their conclusions.

We appreciate the positive comments and have tried to address all concerns from this reviewer to improve the quality of this manuscript.

1- In Figure 1D the signals for H4K12ac seem to overlap only with the brightest CENP-A spots but not with less intense ones. Do the authors have an explanation for this? A control staining with an antibody against a mark that is not enriched at centromeres (e.g. H4K16ac) would be helpful.

Review #2 also asked images of Figure 1D and we performed immunostaining with anti-H4K12 antibody in wild-type HeLa cells (not expressing exogenous GFP-CENP-A). As this reviewer suggested, it is easy to detect H4K12ac signals on centromeres with bright CENP-A signals (Figure 1D). However, some H4K12ac signals were not strong even on centromeres with bright CENP-A signals (new Supplementary Figure 1C). As interphase chromatin forms complicated 3D structure, we interpret that antibodies may easily access some centromeres, which show bright signals. We added this statement in the revised text. In addition, we did not detect any specific signals for H4K16ac even in cells expressing CENP-A-GFP (see below).

2- *To confirm that the observed reduction in CENP-A signal upon RbAp48 depletion is directly connected to failed CENP-A incorporation and not due to an arrest of cells in S- or G2 phase given that RbAp48 contributes to the activity of several different protein complexes, the authors should show cell cycle profiles of these cells.*

We published a detailed characterization of RbAp48-deficient cells previously (Satrimafitrah et al., Chromosome Res., 2016). We measured the FACS profile in this paper (Figure 3) and did not observe a strong cell arrest at G1 or S phase. We describe this point in the revised version, and thank the reviewer for highlighting the need for this important control in this experiment.

3- *In Figure 4 it is shown that the Y32H mutation of RbAp48 reduces binding to CENP-A. However, the authors do not demonstrate that this mutation does not interfere with RbAp48's binding to other complexes. If such proof cannot be obtained, Fig. 4H, I and J provide no substantial additional information to what has already been shown before and could be removed to Supplementary Data.*

As this reviewer suggested, we moved Figure 4H, I, and J to new Supplementary Figure 4D, E, and F, respectively.

4- *As pointed out above and acknowledged by the authors, RbAp48 is a very versatile protein participating in many complexes that are also involved in gene regulation. Therefore, the authors should show that different HJURP expression levels are not the reason for the observed differences in HJURP localization observed in Fig. 5. They do show decreased CENP-A levels in whole cell extracts of RbAp48 OFF cells which is explained by decreased incorporation into chromatin. However, also in this case it needs to be made sure that expression is not affected.*

We agree with this comment and we performed Western blot analysis to examine HJURP level in RbAp48-deficient cells (new Supplementary Figure 5F). As shown in the Supplementary Figure 5F, expression level of HJURP was not changed in RbAp48-deficient cells.

5- *The effects of H4 K5 and K12 mutation on CENP-A mislocalization are interesting. However, the authors do not discuss the possibility that this effect could be secondary to an overall perturbed chromatin structure. In fact, the ChIP data show a decrease of CENP-A signal outside the centromere upon expression of wild-type H4, which can be acetylated and is present in the cells at double dose (shown in Figure S6A). Thus, problems with global efficient reassembly of chromatin during S-phase in cell expressing the acetylation-mutant H4 might promote the previously observed propensity of CENP-A incorporation into extracentromeric chromatin (e.g. Moreno-Moreno et al., NAR 2006).*

We appreciate this comment. Moreno-Moreno et al. proposed that mis-localized CENP-A is degraded by a proteasome-mediated pathway in Drosophila cells. As this reviewer suggested, double dose of H4 decreased amount of non-centromere CENP-A (Figure 6C). Therefore, it is possible that H4 acetylation may be related to CENP-A degradation. We added this point in the revised text.

6. *Fig. 2B: Is the figure mislabeled (it says IP with anti FLAG) or did the authors not use tandem IPs to generate prenucleosomal CENP-A extracts as stated in 2A? Also,*

why are there no HJURP and CENP-A signals in WCE and Input in 2B, when they can readily be detected in all other blots? When solubilizing chromatin with MNase digestion, usually the supernatant after digestion is used. Why did the authors use the pellet in this case (Fig. 1A)?

Figure 2B in previous version was not correct labeling, and we apologize for the error. We performed sequential IP experiments and have corrected labeling.

HJURP and CENP-A stain weakly by western blot in our hands, and it was hard to detect these bands if we normalized intensities for other antibodies. However, if the blot is overexposed, it is possible to detect these bands. We added the long-exposed blot in new Supplementary Figure 2A.

We usually perform MNase digestion at low salt concentration (90mM NaCl) and then increase salt concentration after MNase digestion. Therefore, the fraction we used is solubilized chromatin sample. We explained this in the Figure 2A legend in the revised version.

Finally, the way the manuscript is written it tends to oversell the findings sometimes. For example, in lines 87-88 it says that H4K5ac and K12 ac are "strikingly,highly enriched..", when the ChIP traces do not suggest "high" enrichment. Also, acetylation assays using Hat1 are described on page 5. On page 6 in a new paragraph, the question of which enzyme acetylates H4 K5 and K12 is raised again, in which the Mis18 complex is named as a possible candidate, even though there is no reason to suspect that this complex acts as a Hat enzyme. The authors should streamline these passages.

We appreciate this comment and tried to remove "overstatement" found in previous version, and have streamlined the results section.

We agree with the reviewer that there is no data to implicate the Mis18 complex in direct acetylation of histones. Because modification by Hat1/RbAp46/48 is the most parsimonious explanation, we have removed the statement concerning Mis18 and have revised the text.

Reviewer #2

This manuscript by Shang et al uncovers a role for the RbAp48 histone chaperone in CENP-A assembly. The first indication of such a role dates back to 2004 and its role has been enigmatic since. This paper sheds the first clear mechanistic light on its function at the centromere and as such this is of great interest.

The authors find that chicken DT40 centromeres are enriched for H4K5 and K12 acetylation coinciding with CENP-A. They find H4K12ac to be enriched at the centromere by microscopy. These modifications are enriched predominantly on prenucleosomal CENP-A.

Prenucleosomal CENP-A interacts with RbAp46/48 and HAT-1 as previously observed but not detectably with H3 (in the absence of H4). The HAT-1 interaction with CENP-A/H4 is RbAp46 dependent. Within the prenucleosomal complex H4K5 and 12 are specifically acetylated.

In cells, in the absence of RbAp48, HJURP is lost from the centromere but only if HJURP can interact with CENP-A. HJURP mutants that lack the ability to bind CENP-A localize normally in the absence of RbAp48. This result is somewhat puzzling

and suggest that CENP-A/H4 binding can interfere with HJURP targeting to the centromere, unless bound by RbAp48 and or acetylated on H4. Finally and importantly, Acetyl mimicking mutations on H4 when force expressed in cells can rescue CENP-A targeting to centromeres.

Overall, this paper makes a compelling case on the role of RbAp48 in centromere assembly in DT40 cells. The authors present a comprehensive set of experiments that are consistent with a model where RbAp48 brings HAT-1 to the CENP-A/H4 prenucleosomal complex to acetylate H4 at K5 and k12 that is required for subsequent HJURP mediated deposition of CENP-A/H4 at the centromere. This finding is certainly a valuable contribution to the field and should be published.

We appreciate the reviewer's positive comments.

However, I disagree that these acetylation events should be labelled as "epigenetic marks" for centromere maintenance. H4K5Ac and H4K25Ac are very general modifications that occur on virtually all H4 and is required for general chromatin assembly. I therefore strongly disagree with calling these modifications "epigenetic marks for centromere maintenance". They are neither marks, nor specific for the centromere. They are simply required, just as e.g. the ribosome is required for centromere maintenance. What the authors have done is to demonstrate their requirement and elucidated the role of RbAp48 in this process. There is no need to invoke terminology that muddles their findings. The title and text should be modified to better match their data.

We appreciate this comment and changed the title to "Acetylation of histone H4 Lysine 5 and 12 is required for CENP-A deposition into centromeres". In the text we did not refer these modification as "epigenetic marks".

Other key issues to be addressed before publication:

Figure 1D:

Why is ectopic CENP-A GFP overexpression performed here? It is possible that H4K12Ac enrichment at the centromere is induced by CENP-A overexpression. Why not use antibody staining against endogenous centromere proteins to mark centromere position?

We performed immunostaining with H4K12ac in HeLa cells not expressing exogenous CENP-A-GFP (new Supplementary Figure 1C) and show that there is overlapping signal with endogenous CENP-A.

Figure 3G:

Ref 27 showed that specificity between Mis16 (the RbAp homolog in fission yeast) and CENP-A comes from Mis16 binding Scm3, the fission yeast HJURP homolog. In the reactions in figure 3G there is no HJURP. How do the authors explain the specific interaction of RbAp46 to CENP-A and the apparent species difference?

We detect a reproducible and robust direct interaction between CENP-A and RbAp46/48 in the absence of HJURP (Scm3 homologue). The paper of Ref 27 from Uhn-Soo Cho's group describes an interaction between Mis16 (RbAp46/48) and H4 in *S. pombe*. The analogous interaction has been shown by Alain Verrault and Bruce Stillman for the vertebrate homologs. In the work from the Cho lab they show that Scm3 can bind directly to Mis16 and can do so

while the Mis16 protein is also bound to H4, indicating that the binding sites for H4 and Scm3 are distinct. In that paper, however, a direct interaction between Cnp1 (CENP-A) and Mis16 (RbAp46/48) was never tested. The model those authors propose is a reasonable one, which says that Mis16 (RbAp46/48) has specificity for CENP-A because it is also interacting with Scm3, however none of the data in that paper rule out an interaction between Mis16 (RbAp46/48) and Cnp1 (CENP-A).

Our data demonstrates that interaction for the vertebrate proteins. Because *S. pombe* and vertebrate CENP-A are highly diverged in the N-terminus, it is unclear whether this interaction will occur in *S. pombe* as it does in humans, but as our data stands there is no conflict between the data, our results add to the biochemical results studying the *S. pombe* protein interaction between RbAp46/48 and H4.

Line 165:

"Another candidate is the Mis18 complex, as tricostatin A, a histone deacetylase inhibitor, suppresses the phenotype of Mis18 α -knockdown cells".

Masumoto's lab has recently identified a relevant HAT (KAT7) that may be responsible. This work should be cited and discussed here.

We cited the Masumoto paper in the revised version. In addition, we tested whether Kat7 may be involved in H4K5 and K12 acetylation in centromeres based on Kat7 KO cells and ChIP-seq analysis. We found that levels H4K5 and K12 acetylation in centromeres were not changed in Kat7 KO cells. We added these data into new Supplementary Figure 2G in the revised version.

Figure 3H, By mutating K12, most of HAT1 mediated acetylation is lost, suggesting that K12 is the main target for HAT1. This does not exclude the possibility that K5 is not a main target as well. For instance, if K5 and K12 are interdependent then lack of K5 acetylation may prevent K12 acetylation and visa versa. To clearly identify K12 as the main target mutant K5 should be tested as well in this assay.

We have provided additional data with H4_K5R_K12R or H4_K5Q_K12Q double mutants (new Supplementary Figure 2F). Although K12ac is main target for Hat1, we are able to detect a faint acetylation signal in the H4_K12R or K12Q mutants as a substrate. This is consistent with the observed preference for K12 over K5 by Hat1. Kinetic and structural analysis of K5 and K12 acetylation have demonstrated that K12 is the preferred substrate for acetylation and that this residue is quantitatively mono-acetylated on K12 before K5 acetylation begins (Wu et al. PNAS 2012). In addition, previous biochemical studies have shown that pre-acetylating either K5 or K12 does not prevent the acetylation of the other residue by Hat1 (Benson et al. JBC 2007, Makowski et al. JBC 2001). It has also been shown that once K12 is acetylated the affinity of the histone tail for the Hat1 enzyme drops 20 fold (Wu et al. PNAS 2012). Thus we favor a model in which K12 is the preferred substrate for Hat1 and is preferentially acetylated while K5 is modified only after K12 modification is complete. It is worth noting, however that our studies are done with the K5 and K12 double mutants so that even if K5 acetylation plays a more important role in vivo than in vitro we should still abrogate its effect in our assays.

Figure 4F

The Y32H mutant of RbAp48 has impaired CENP-A binding but normal HJURP binding. The authors observe that HJURP targeting to centromeres is prevented in RbAp48 mutants presumably due to a lack of acetylation of H4 which they suggest interferes with HJURP targeting to centromeres.

Alternatively, RbAp48 is required for HJURP to bind to centromeres through a direct interaction as has been suggested by the work in fission yeast (Ref27). If so, this would predict that in the RbAp48 mutant at Y32H, HJURP would still target normally to centromeres, as in this case no unacetylated CENP-A/H4 is present in the HJURP/RbAP48 complex. This should be tested in order to distinguish between these possibilities.

As suggested, we directly tested whether HJURP localizes to centromeres in cells expressing Y32H RbAp48 mutant. New Supplementary Figure 4G indicates that HJURP does not localize centromeres properly, suggesting that RbAp48 may not directly involved in HJURP localization into centromere. As we also demonstrate that N terminal truncated HJURP can localize centromere in the absence of RbAp48 (Figure 5E). This also suggests that RbAp48 may not be directly involved in centromere localization of HJURP. We thank the reviewer for this helpful suggestion.

Figure 7B. The loss of non-centromeric CENP-A is shown, why not quantify directly the rescue of centromeric CENP-A levels? It is critical to directly demonstrate and quantify to what extent K5 and K12 mutations rescue the CENP-A loading defect. The authors present a rather indirect measure.

Perhaps mutant H4 interferes with tetracycline mediated repression of RbAp48. Although unlikely, this would be a trivial explanation for the rescue. Did the authors check this RbAp48 loss in the OFF condition in the presence of mutant H4?

In RbAp48-deficient cells CENP-A deposition is reduced (42% compared with control cells). If H4Q5Q12 was expressed, defect of new CENP-A deposition into centromere is weakly rescued (49% compared with control cells). However, mis-localization of CENP-A is reduced significantly in RbAp48-deficient cells expressing H4Q5Q12. While we agree with this comment that it is odd to show only non-centromeric CENP-A, it is most important to show ratio of centromeric vs non-centromeric CENP-A. Therefore, we show intensity ratio of centromeric vs non-centromeric CENP-A in the revised version (Figure 7B).

In addition, according to suggestion of this reviewer, we demonstrate that H4 mutants does not interfere tetracycline mediated repression of RbAp48 (new Supplementary Figure 6C).

Reviewers' comments:

Reviewer #1 (Remarks to the Author):

In the revised version of their paper, Shang et al. addressed most of my previous issues in an adequate fashion. However, the claim that RbAp48 reduction is directly responsible for reduced CENP-A incorporation (page 8 of the revised manuscript) remains worrisome as it is not fully supported by the shown and previously published results.

In fact, in their earlier paper (PMID:26667624), the authors show that RbAp48 depletion results in clear defects in S-phase progression (Figure 2), which obviously needs to be taken into account, when interpreting reduced CENP-A levels.

It is therefore not sufficient to state “As RbAp48 OFF DT40 cells proceed through the cell cycle 20, these data suggest that RbAp48 deletion directly causes a defect in CENP-A deposition during G1” as the authors do on page 8 of the revised manuscript. They need to clearly state the time after tetracyclin addition to the cells, at which the measurements in this paper were taken because this appears to greatly affect progression through S-phase (PMID:26667624), and they need to discuss the potential problems with a delayed and compromised S-phase for the observed CENP-A-related phenotypes.

Reviewer #2 (Remarks to the Author):

In this revised version, the authors have adequately responded to all my initial concerns. As per my request, they have added additional data and controls to address all my points.

Important improvements include:

Removal of claims that the H4 acetylation constitutes an epigenetic mark for centromere identity.

Demonstrating centromere localized H4 acetylation in cell expressing endogenous levels of CENP-A.

Providing additional data and citations on the specificity of HAT1/RbAp48 for K12 over K5 of H4.

With this new data and text revisions, I have no further criticisms that should prevent publication.

Response to Reviewers Shang et al. (Nature Communications manuscript NCOMMS-16-14845-A)

Reviewer #1

In the revised version of their paper, Shang et al. addressed most of my previous issues in an adequate fashion.

We appreciate the positive comments and have further tried to address another concern from this reviewer to improve the quality of this manuscript.

However, the claim that RbAp48 reduction is directly responsible for reduced CENP-A incorporation (page 8 of the revised manuscript) remains worrisome as it is not fully supported by the shown and previously published results.

In fact, in their earlier paper (PMID:26667624), the authors show that RbAp48 depletion results in clear defects in S-phase progression (Figure 2), which obviously needs to be taken into account, when interpreting reduced CENP-A levels.

It is therefore not sufficient to state “As RbAp48 OFF DT40 cells proceed through the cell cycle 20, these data suggest that RbAp48 deletion directly causes a defect in CENP-A deposition during G1” as the authors do on page 8 of the revised manuscript. They need to clearly state the time after tetracycline addition to the cells, at which the measurements in this paper were taken because this appears to greatly affect progression through S-phase (PMID:26667624), and they need to discuss the potential problems with a delayed and compromised S-phase for the observed CENP-A-related phenotypes.

We agree with this comment and changed text in line with this comment. We actually performed the SNAP assay during 4 h from 44 h to 48 h after tetracycline addition. As this reviewer pointed out, S-phase defect occurs in RbAp48-deficient cells. However, during the time period from 44 h to 48 h after tetracycline addition, cells are still growing and significant numbers of G1 cells were observed (After 48h cell growth are completely stopped). Therefore, we performed this experiment in this time period. In the revised version, we clearly state time point at which we did the assay and discussed the potential problem.

Reviewer #2

In this revised version, the authors have adequately responded to all my initial concerns. As per my request, they have added additional data and controls to address all my points.

Important improvements include:

Removal of claims that the H4 acetylation constitutes an epigenetic mark for centromere identity.

Demonstrating centromere localized H4 acetylation in cell expressing endogenous levels of CENP-A.

Providing additional data and citations on the specificity of HAT1/RbAp48 for K12 over K5 of H4.

We appreciate the reviewer's positive comments. We hope that this version is suitable for publication.

Reviewers' Comments:

Reviewer #1 (Remarks to the Author):

In the revised version the authors clarify the experimental conditions and clearly state the potential cell-cycle related problems that may affect interpretation of the results.

I therefore have no more reservations about the support of this interesting work.